# Simultaneous Multi-Robot Motion Planning with Projected Diffusion Models

**Jinhao Liang** [1]   **Jacob K. Christopher** [1]   **Sven Koenig** [2]   **Ferdinando Fioretto** [1]

## Abstract

Recent advances in diffusion models hold significant potential in robotics, enabling the generation of diverse and smooth trajectories directly from raw representations of the environment. Despite this promise, applying diffusion models to motion planning remains challenging due to their difficulty in enforcing critical constraints, such as collision avoidance and kinematic feasibility. These limitations become even more pronounced in Multi-Robot Motion Planning (MRMP), where multiple robots must coordinate in shared spaces. To address these challenges, this work proposes **S**imultaneous **M**RMP **D**iffusion (SMD), a novel approach integrating constrained optimization into the diffusion sampling process to produce collision-free, kinematically feasible trajectories. Additionally, the paper introduces a comprehensive MRMP benchmark to evaluate trajectory planning algorithms across scenarios with varying robot densities, obstacle complexities, and motion constraints. Experimental results show SMD consistently outperforms classical and other learning-based motion planners, achieving higher success rates and efficiency in complex multi-robot environments. The code and implementation are available at `https://github.com/RAISELab-atUVA/Diffusion-MRMP`.

## 1. Introduction

Multi-Robot Motion Planning (MRMP) is a fundamental problem in robotics and autonomous systems, where the goal is to compute collision-free paths for multiple robots navigating shared environments (Luo et al., 2024; Shaoul et al., 2025). MRMP has widespread applications, from autonomous vehicles to warehouse logistics and search-and-rescue, where robots must operate reliably in complex and highly constrained settings. Despite its importance, solving MRMP in real-world scenarios remains challenging due to the need to plan trajectories in continuous spaces and the unstructured nature of their inputs. These inputs often lack the structured representations needed by classical algorithms, which require implicit representation of both robot state spaces and obstacles in configuration spaces (LaValle, 2006; Ichter et al., 2018; Luo et al., 2024). Consequently, classical approaches struggle to operate effectively in these high-dimensional environments (Wang et al., 2021; Teng et al., 2023). Specifically, sampling-based planners often yield non-smooth paths due to their reliance on discrete connectivity between sampled configurations (Carvalho et al., 2024). Optimization-based methods suffer from similar issues, as they require computationally expensive trajectory refinements to enforce smoothness and feasibility (Ichnowski et al., 2020).

To address these challenges, learning-based methods offer a promising alternative by leveraging data-driven priors to handle unstructured environments. Recently, diffusion models, a class of generative models originally developed for image and signal processing tasks (Song & Ermon, 2019; Ho et al., 2020), have shown promise for single-robot motion planning (Carvalho et al., 2023; Christopher et al., 2024). However, extending diffusion models to MRMP presents new challenges: while these models effectively produce diverse trajectories, they fail to enforce hard constraints such as collision avoidance and kinematic feasibility, a limitation that becomes increasingly significant in MRMP.

Existing diffusion-based approaches attempt to address these constraints through gradient-based guidance or rejection sampling (Okumura et al., 2022b; Carvalho et al., 2023). However, gradient-based methods are prone to fail in such scenarios due to the non-convex nature of collision avoidance constraints, complicating trajectory correction. Rejection sampling methods, conversely, suffer from inefficiencies, discarding large portions of generated trajectories and struggling to generate feasible solutions in cluttered environments (Carvalho et al., 2024; Christopher et al., 2024).

To address these limitations, this paper introduces *Simultaneous MRMP Diffusion* (SMD), a novel approach that integrates constrained optimization directly into the diffusion

[1]Department of Computer Science, University of Virginia, Charlottesville, VA 22903, USA [2]Department of Computer Science, University of California, Irvine, CA 92697, USA. Correspondence to: Ferdinando Fioretto <fioretto@virginia.edu>.

*Proceedings of the 42$^{nd}$ International Conference on Machine Learning*, Vancouver, Canada. PMLR 267, 2025. Copyright 2025 by the author(s).

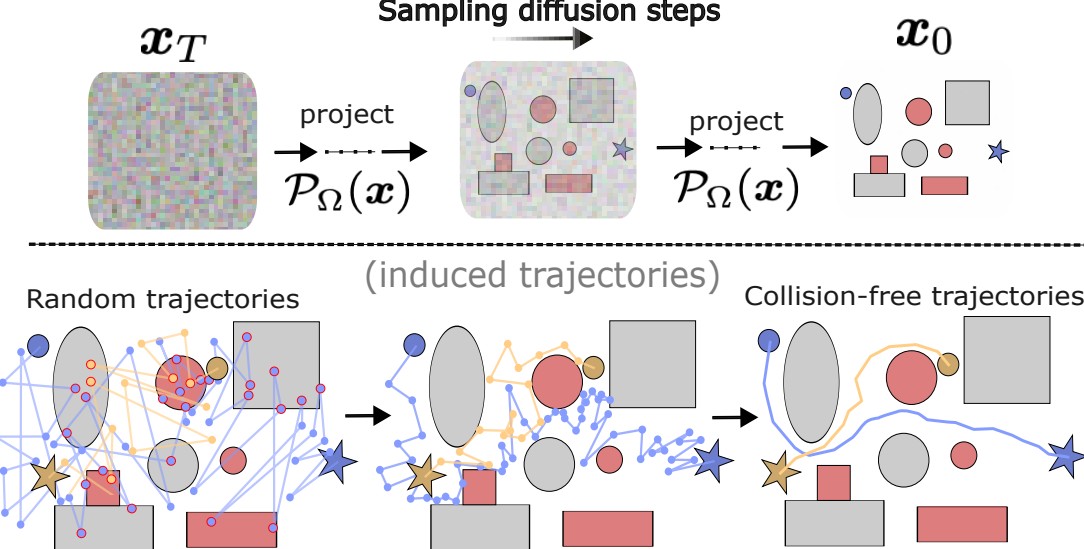

Figure 1. SMD incorporates a projection operator to enforce constraints within the diffusion process. During sampling, the projection operator iteratively corrects trajectories by mapping them to the nearest feasible points, resulting in final collision-free paths. **The infeasible paths are marked with red circles and only occur in the initial random trajectory.** In the experiments, the red objects appear at sampling time only, and the robots move from their start ● to their goal positions ⋆.

sampling process to generate feasible multi-robot trajectories. This work reformulates MRMP within a constrained diffusion framework, where diffusion-based trajectory generation is guided by an Lagrangian-dual based method. This approach enables diffusion models to satisfy collision avoidance and kinematic constraints without post-hoc filtering, making them applicable even in high-density, unstructured environments. Unlike prior methods, which degrade significantly in cluttered scenarios, our framework maintains feasibility even as the number of robots and obstacles increases. The overall scheme is illustrated in Figure 1.

**Contributions.** The paper makes the following contributions: **(1)** It introduces Simultaneous MRMP Diffusion (SMD), which formulates multi-robot trajectory generation as a constrained diffusion process, ensuring collision-free and kinematically feasible motion plans. **(2)** It integrates constrained optimization directly into the sampling process of diffusion models, enabling the direct embedding of constraints within the trajectory generation pipeline. **(3)** It develops a Lagrangian-dual based method to reformulate the MRMP problem, dramatically improving the efficiency with these critical constraints can be satisfied during diffusion sampling, both theoretically and empirically. **(4)** Finally, it introduces the first benchmark for MRMP evaluation, featuring complex input maps and diverse scenarios. Our approach demonstrates significant improvements over competing methods, particularly in environments with dense obstacles and unstructured configurations.

## 2. Related Work

**Motion Planning.** Motion planning computes a feasible path for a robot to move from its start to its goal state while avoiding obstacle collisions. Sampling-based algorithms, such as Probabilistic Roadmaps (Kavraki et al., 1996) and Rapidly-Exploring Random Trees (RRTs) (LaValle, 1998) have been widely studied for this task. These methods guarantee probabilistic completeness but face significant scalability issues, especifically in multi-robot motion planning (MRMP) as the configuration space of MRMP scales significantly. Alternatively, MRMP can be formulated as a constrained optimization problem, solving by optimization methods such as gradient optimization techniques (Ratliff et al., 2009) and sequential convex programming (Augugliaro et al., 2012; Chen et al., 2015). However, these approaches encounter difficulties when taking into account changes in acceleration (Ichnowski et al., 2020) and are usually too conservative to find any solution, even if one exists (Chen et al., 2015). Other lines of research focus on a discretized version of this problem, known as multi-agent path finding. Specifically, multi-agent path finding reduces the configuration space by discretizing both time and space into steps and grids, respectively (Stern et al., 2019). This simplification has enabled the development of efficient search-based algorithms (Li et al., 2019; 2021; Okumura et al., 2022a). However, these assumptions create a disconnect from real-world applications due to their oversimplification (Shaoul et al., 2025). Unlike these approaches, this paper develops SMD, a diffusion-based MRMP frame-

work that directly learns the distribution of collision-free paths and generates feasible solutions for multiple robots simultaneously in complex environments.

**Motion Planning with Generative Models.** Recent advancements in generative models have opened new avenues for solving motion planning problems by learning complex, high-dimensional distributions of feasible paths. For example, Okumura et al. (2022b) employed a conditional variational autoencoder to predict cooperative timed roadmaps. Following the introduction of diffusion models into motion planning by Janner et al. (2022), subsequent research has largely focused on using gradient-based methods to guide the outputs of diffusion models toward feasible solutions (Carvalho et al., 2023; Luo et al., 2024; Saha et al., 2024; Ubukata et al., 2024; Carvalho et al., 2024; Naderiparizi et al., 2025). These works design tailored cost functions to guide the generation of diffusion models during the sampling process. Although these approaches leverage the ability of diffusion models to generate diverse trajectories, they are unable to ensure constraint satisfaction in scenarios with complex environments and multiple robots. In fact, most existing work has primarily addressed single-robot motion planning, leaving multi-robot motion planning underexplored. Recently, Shaoul et al. (2025) combined diffusion models with search-based multi-agent path finding algorithms, where the diffusion models generate trajectories for a single robot, and a search-based algorithm determines the final trajectories for multiple robots. While this is a substantial contribution towards MRMP, it does not ensure the feasibility of the trajectories generated by diffusion models, particularly in environments with dense obstacles or complex robot interactions. In contrast, our proposed SMD directly integrates optimization techniques into the diffusion process, enabling the generation of feasible MRMP trajectories even in scenarios with a significant number of robots and obstacles.

## 3. Preliminaries

**Score-based Diffusion Models.** Generative diffusion models (Sohl-Dickstein et al., 2015; Ho et al., 2020) produce high-fidelity complex data operating in two phases: A *forward step*, that gradually introduce noise to clean data samples, followed by a *reverse denoising step*, in which a deep neural network is trained to iteratively remove noise. This forward process defines a Markov chain $\{\boldsymbol{x}_t\}_{t=0}^{T}$, with initial sample $\boldsymbol{x}_0 \sim p(\boldsymbol{x}_0)$ and each transition $q(\boldsymbol{x}_t|\boldsymbol{x}_{t-1})$ realized by adding Gaussian noise according to a variance schedule $\alpha_t$. In *score-based diffusion* (Song & Ermon, 2019; Song et al., 2020), the reverse process relies on a learned score function $s_\theta(\boldsymbol{x}_t, t) = \nabla_{\boldsymbol{x}_t} \log p(\boldsymbol{x}_t)$ that approximates the gradient of the log probability density of the noisy data.

This score function is trained to minimize

$$\min_{\theta} \underset{\substack{t \sim [1,T], p(\boldsymbol{x}_0), \\ q(\boldsymbol{x}_t|\boldsymbol{x}_0)}}{\mathbb{E}} (1-\alpha_t) \left[ \|s_\theta(\boldsymbol{x}_t, t) - \nabla_{\boldsymbol{x}_t} \log q(\boldsymbol{x}_t|\boldsymbol{x}_0)\|^2 \right],$$

and is then used to iteratively "denoise" random noise samples back into data-like samples. This is also known as the *sampling* phase.

**Multi-Robot Motion Planning.** Multi-Robot Motion Planning (MRMP) involves computing collision-free trajectories for multiple robots navigating a shared environment from designated start positions to goal states. Consider a set of $N_a$ robots $\mathcal{A} = \{a_1, a_2, \ldots, a_{N_a}\}$ operating in a continuous workspace. Each robot $a_i$ is modeled as a sphere with radius $r_i > 0$ and has a trajectory over $H$ time steps denoted by $\boldsymbol{\pi}_i = [\pi_i^1, \pi_i^2, \ldots, \pi_i^H]$, where $\pi_i^h = (x_i^h, y_i^h) \in \mathbb{R}^2$ represents the robot's states at time $h$. For each robot, their start and target states are defined, respectively by sets $\mathbf{B} = [b_1, b_2, \ldots, b_{N_a}]$ and $\mathbf{E} = [e_1, e_2, \ldots, e_{N_a}]$. The robots must navigate around $N_o$ obstacles $\mathcal{O} = \{o_1, \ldots, o_{N_o}\}$ while adhering to kinematic constraints such as velocity and acceleration limits.

The objective is to compute a feasible set of trajectories $\mathbf{\Pi} = \{\boldsymbol{\pi}_1, \boldsymbol{\pi}_2, \ldots, \boldsymbol{\pi}_{N_a}\}$, that minimizes a predefined cost function while ensuring feasibility to environmental constraints and inter-robot collision avoidance. Formally,

$$\min_{\mathbf{\Pi}} \quad \mathcal{J}(\mathbf{\Pi}) \tag{1a}$$

$$\text{s.t.} \quad \mathbf{\Pi} \subseteq \Omega_{\text{obs}}, \tag{1b}$$

$$\pi_i^1 = b_i, \quad \forall i \in [N_a], \tag{1c}$$

$$\pi_i^H = e_i, \quad \forall i \in [N_a], \tag{1d}$$

$$\text{Kinematic constraints on } \mathbf{\Pi}, \tag{1e}$$

$$\text{Collision avoidance between robots in } \mathbf{\Pi}, \tag{1f}$$

where $\mathcal{J} : \mathbb{R}^{N_a \times H \times 2} \to \mathbb{R}^+$ represents the cost function, which may include objectives such as total travel time or energy consumption, and $\Omega_{\text{obs}}$ is the feasible region, excluding obstacle-occupied space. Constraints (1b) ensure that robots avoid obstacles, (1c) and (1d) ensure that each robot starts at its initial position and reaches its target position. (1e) enforce kinematic limits, defined by a maximum reachable velocity, and (1f) ensures collision avoidance, maintaining a minimum Euclidean separation between robots at all time steps. Subsequently, we denote constraint set (1b)–(1f) with $\Omega$.

The MRMP problem poses significant challenges due to the difficulty of representing and navigating unstructured, high-dimensional configuration spaces and coordinating dynamic interactions among multiple robots in continuous environments (Stern et al., 2019; Shaoul et al., 2025).

# 4. Simultaneous MRMP Diffusion

To address these challenges, this section introduces Simultaneous MRMP Diffusion (SMD), a diffusion-based method for MRMP. We start by introducing the concept of repeated projections, which constitutes a core building block of SMD. Then, the section discusses how to integrate the multi-robot motion planning constraints into the diffusion framework and, finally, how to enable effective sampling in complex, high-dimensional MRMP tasks. A theoretical analysis is also provided to establish the feasibility guarantees of SMD.

## 4.1. Repeated Projections.

Incorporating constrained optimization techniques into generative models has been an important direction in ensuring feasibility in structured domains. Diffusion models inherently use a variant of *Langevin Monte Carlo sampling*, known as *Stochastic Gradient Langevin Dynamics* (SGLD), for their denoising sampling process. SGLD introduces an additional stochastic perturbation to gradient-based updates, resulting in a non-deterministic version of natural gradient descent. This allows it to from getting trapped in local minima and enables diverse sampling trajectories (Welling & Teh, 2011). Given this perspective, the diffusion sampling process can be viewed as an iterative unconstrained optimization problem aimed at maximizing the log-likelihood of the true data distribution.

Building upon this understanding from Christopher et al. (2024), we can extend the standard diffusion process by incorporating a feasible region $\Omega$:

$$\min_{\mathbf{x}_T,\ldots,\mathbf{x}_1} \sum_{t=T,\ldots,1} -\log q(\mathbf{x}_t|\mathbf{x}_0) \quad (2a)$$

$$\text{s.t.} \quad \mathbf{x}_T,\ldots,\mathbf{x}_0 \in \Omega. \quad (2b)$$

To enforce these constraints during the generative process, it is possible to modify the standard Stochastic Gradient Langevin Dynamics (SGLD) update rule by introducing a *projection step* after each iteration:

$$\mathbf{x}_t^{i+1} = \mathcal{P}_\Omega \underbrace{\left(\mathbf{x}_t^i + \gamma_t \nabla_{\mathbf{x}_t^i} \log q(\mathbf{x}_t|\mathbf{x}_0) + \sqrt{2\gamma_t}\mathbf{z}\right)}_{\text{classic reverse process}}, \quad (3)$$

where $\mathbf{z}$ is standard normal, $\gamma_t > 0$ is the step size, $\nabla_{\mathbf{x}_t^i} \log q(\mathbf{x}_t|\mathbf{x}_0)$ is approximated by the learned score function $\mathbf{s}_\theta(\mathbf{x}_t, t)$, $\Omega$ is the set of constraints, and $\mathcal{P}_\Omega(\cdot)$ is a projection onto $\Omega$:

$$\mathcal{P}_\Omega(\mathbf{x}) = \arg\min_{\mathbf{x}' \in \Omega} \|\mathbf{x} - \mathbf{x}'\|^2. \quad (4)$$

At each time step $t$, starting from $\mathbf{x}_t^0$, the process performs $M$ iterations of SGLD.

A key theoretical advantage of this approach is that it retains the convergence guarantees of Langevin-based sampling. Specifically, this constrained diffusion process *converges to an "almost-minimizer" of the objective function* (2a), where the approximation quality is bounded by the noise term $\mathbf{z}$ and the Langevin step size $\gamma_t$ (Xu et al., 2018).

Importantly, as shown in (Christopher et al., 2024), this process guarantees that the generated outputs satisfy constraints from convex sets under mild conditions.

## 4.2. Collision-free Projection Mechanism

While the application of repeated projections provides a useful primitive to steer samples generated by diffusion models to satisfy relevant constraints, projecting onto the space of collision-free and kinematically viable trajectories presents a critical challenge due to the nonconvex nature of these constraints and the high dimensionality of the problem. These issues make the standard application of repeated projections ineffective in solving even simple MRMP instances. The following first presents the projection mechanism for SMD, which aims to generate feasible trajectories for all robots in the problem.

Note that the feasible region $\Omega$ for the MRMP problem can be represented by distinguishing between convex and nonconvex constraints. This distinction will be useful later to provide an accelerated version of the projection operator introduced above.

**Convex Constraints.** First, note that each robot's trajectory must begin and end at its designated start and goal locations, as enforced by Constraints (1c) and (1d). Additionally, robots must respect maximum velocity limits between consecutive time steps:

$$\left(\pi_i^h - \pi_i^{h-1}\right)^2 \leq \left(v_i^{\max}\Delta t\right)^2, \ \forall i \in [N_a], h \in \{2,\ldots,H\}, \quad (5)$$

where $v_i^{\max}$ represents the maximum allowable velocity for robot $a_i$, and $\Delta t$ is the time interval between steps. Together, these constraints define a *convex set*:

$$\Omega_c = \left\{\mathbf{\Pi} \in \mathbb{R}^{N_a \times H \times 2} \Big| \text{Constr. (1c), (1d), and (5) hold}\right\}.$$

**Nonconvex Constraints.** To ensure collision-avoidance the following nonconvex constraints must also be imposed:

$$(\pi_i^h - \pi_j^h)^2 \geq (R^a)^2, \forall i,j, \ i \neq j \in [N_a], h \in [H], \quad (6)$$

where $R^a$ denotes the minimum distance between robots at any time. The above states that any two robots should be far enough from each other at any point in the trajectory.

Similarly, the following constraints are imposed to avoid collisions between each robot and static obstacles:

$$(\pi_i^h - o_j)^2 \geq (R^o)^2, \forall i \in [N_a], \forall j \in [N_o], h \in [H], \quad (7)$$

**Algorithm 1** Diffusion Sampling Process in SMD

1: **Input:** Gaussian Noise $\mathbf{x}_T^0$
2: **for** $t = T \to 1$ **do**
3:     **Initialize** $\gamma_t$
4:     **for** $i = 1 \to M$ **do**
5:         Sample $\mathbf{z} \sim \mathcal{N}(\mathbf{0}, \mathbf{I})$
6:         Compute $\mathbf{g} \leftarrow s_\theta(\mathbf{x}_t^{i-1}, t)$
7:         Update $\mathbf{x}_t^i \leftarrow \mathcal{P}_\Omega(\mathbf{x}_t^{i-1} + \gamma_t \mathbf{g} + \sqrt{2\gamma_t}\mathbf{z})$
8:     **end for**
9:     $\mathbf{x}_{t-1}^0 \leftarrow \mathbf{x}_t^M$
10: **end for**
11: **Output:** $\mathbf{x}_0^0$

where $o_j$ is the position of obstacle $j$ and $R^o$ denotes the minimum distance at which an robot must be to avoid an obstacle. Together, these constraints define a *nonconvex set*:

$$\Omega_n = \left\{ \mathbf{\Pi} \in \mathbb{R}^{N_a \times H \times 2} \Big| \text{Constr. (6), (7), hold} \right\},$$

and the complete feasible set is given by: $\Omega = \Omega_c \cap \Omega_n$.

The detailed algorithm is shown in Algorithm 1. Notably, the SMD framework allows straightforward incorporation of additional constraints, such as acceleration bounds or trajectory smoothness, by introducing these to the feasible set $\Omega$.

### 4.3. Lagrangian Relaxation for Efficient Projections

Although solving $\mathcal{P}_\Omega(\boldsymbol{x})$ can generate feasible trajectories, the nonconvex nature of the constraint set above results in high computational costs. To address this issue, we propose a relaxation of the nonconvex constraints in MRMP (Boyd et al., 2011) to vastly enhance the tractability of projections onto a MRMP feasible set. Following standard practices in constrained optimization, we transform inequality constraints into equality constraints using non-negative auxiliary variables (Kotary & Fioretto, 2024). This simplifies multiplier updates and improves convergence:

$$\mathcal{H}_a : (\pi_i^h - \pi_j^h)^2 - d_{i,j,h}^a - (R^a)^2 = 0, \forall i, j, \; i \neq j, \; \forall h,$$
$$\mathcal{H}_o : (\pi_i^h - o_j)^2 - d_{i,j,h}^o - (R^o)^2 = 0, \forall i, j, \; \forall h,$$

where $d_{i,j,h}^a$ and $d_{i,j,h}^o$ (with vector form $\boldsymbol{d^a}$ and $\boldsymbol{d^o}$, respectively) are positive auxiliary variables. Specifically, $\mathcal{H}_a$ corresponds to the robot collision avoidance constraints and $\mathcal{H}_o$ to the obstacle collision avoidance constraints.

The Lagrangian function of the MRMP problem is thus defined as:

$$\mathcal{L}(\mathbf{\Pi}, \boldsymbol{\nu}_a, \boldsymbol{\nu}_o) = \mathcal{J}(\mathbf{\Pi}) + \boldsymbol{\nu}_a^\top \mathcal{H}_a(\mathbf{\Pi}) + \boldsymbol{\nu}_o^\top \mathcal{H}_o(\mathbf{\Pi}), \quad (9)$$

where $\mathcal{J}(\mathbf{\Pi})$ represents the objective function, which typically finds the nearest feasible point to the input in the projection, as shown in Eq. 4. $\boldsymbol{\nu}_a$ and $\boldsymbol{\nu}_o$ are Lagrangian multipliers, and $\mathcal{H}_a$ and $\mathcal{H}_o$ represent the equality constraints

enforcing inter-robot collision avoidance and obstacle avoidance, respectively. While the standard Lagrangian formulation provides a means of incorporating constraints into the optimization process, it suffers from slow convergence due to the instability of dual variable updates, particularly in nonconvex settings. To mitigate these issues, we adopt the augmented Lagrangian method (Boyd et al., 2011; Kotary et al., 2022) to improve the convergence performance, which introduces additional penalty terms on the constraint residuals:

$$\begin{aligned} \mathcal{L}(\mathbf{\Pi}, \boldsymbol{\nu}_a, \boldsymbol{\nu}_o) = \quad & \mathcal{J}(\mathbf{\Pi}) \\ & + \boldsymbol{\nu}_a^\top \mathcal{H}_a(\mathbf{\Pi}) + \boldsymbol{\nu}_o^\top \mathcal{H}_0(\mathbf{\Pi}) \\ & + \rho_a \|\mathcal{H}_a(\mathbf{\Pi})\|^2 + \rho_o \|\mathcal{H}_o(\mathbf{\Pi})\|^2, \end{aligned}$$

where $\rho_a$ and $\rho_o$ are penalty parameters that control the strength of the constraint enforcement. These penalties improve numerical stability by discouraging constraint violations in early iterations, accelerating convergence.

The corresponding Lagrangian dual function can thus be defined by:

$$\boldsymbol{d}(\boldsymbol{\nu}_a, \boldsymbol{\nu}_o) = \min_{\mathbf{\Pi}} \mathcal{L}(\mathbf{\Pi}, \boldsymbol{\nu}_a, \boldsymbol{\nu}_o).$$

and associated *Lagrangian Dual Problem* is defined as maximizing the dual function:

$$\arg \max_{\boldsymbol{\nu}_a, \boldsymbol{\nu}_o} \boldsymbol{d}(\boldsymbol{\nu}_a, \boldsymbol{\nu}_o) \qquad \text{s.t.} \quad \mathbf{\Pi} \in \Omega_c. \qquad (10)$$

Through weak duality, maximizing (10) provides a lower bound on the optimal objective value of the primal MRMP problem. Even in cases where strong duality does not hold (e.g., as in the application context studied in this work), minimizing the duality gap allows for near-optimal feasible solutions to be obtained. Given the optimal dual variables $(\boldsymbol{\nu}_a^*, \boldsymbol{\nu}_o^*)$, a primal solution $\hat{\mathbf{\Pi}}$ can be obtained by solving:

$$\hat{\mathbf{\Pi}} = \arg \min_{\mathbf{\Pi} \in \Omega_c} \mathcal{L}(\mathbf{\Pi}, \boldsymbol{\nu}_a^*, \boldsymbol{\nu}_o^*).$$

This follows from the stationarity condition, where optimizing the augmented Lagrangian function with fixed multipliers yields the best feasible solution.

Crucially, the dual problem (10) can be solved iteratively, by employing an adaptation of a Dual Ascent method (Boyd et al., 2011):

$$\mathbf{\Pi}^k = \arg \min_{\mathbf{\Pi} \in \Omega_c} \mathcal{L}(\mathbf{\Pi}, \boldsymbol{\nu}_a^k, \boldsymbol{\nu}_o^k), \qquad (12a)$$
$$\boldsymbol{\nu}_a^{k+1} = \boldsymbol{\nu}_a^k + \rho_a^k \mathcal{H}_a(\mathbf{\Pi}^k), \qquad (12b)$$
$$\boldsymbol{\nu}_o^{k+1} = \boldsymbol{\nu}_a^k + \rho_o^k \mathcal{H}_o(\mathbf{\Pi}^k). \qquad (12c)$$

At each iteration, the primal variable $\mathbf{\Pi}^k$ is updated by minimizing the augmented Lagrangian, and the dual variables

**Algorithm 2** Efficient Projection Operator for SMD
___
1: **Input:** Tolerance $\delta_a$ and $\delta_o$, Weight $\rho$, Initial Trajectory $\hat{\Pi}$, scaling factor $\zeta$
2: **while** $\nabla_{\nu_a} < \delta_a$ **and** $\nabla_{\nu_o} < \delta_o$ **do**
3:     $\hat{\nu}_a \leftarrow \mathcal{H}_a(\hat{\Pi}), \hat{\nu}_o \leftarrow \mathcal{H}_o(\hat{\Pi})$
4:     $\hat{\Pi} \leftarrow \arg\min_{\Pi \in \Omega_c} \mathcal{L}(\hat{\Pi}, \nu_a^*, \nu_o^*)$
5:     $\nabla_{\nu_a} \leftarrow \mathcal{H}_a(\hat{\Pi}), \nabla_{\nu_o} \leftarrow \mathcal{H}_o(\hat{\Pi})$
6:     $\rho \leftarrow \zeta \times \rho$
7: **end while**
8: **Output:** $\hat{\Pi}$
___

$\nu_a, \nu_o$ are updated via a gradient ascent step on the constraint residuals. This approach enables the generation of collision-free and kinematic viable trajectories for multiple robots, especially in complex scenarios. The efficient projection process is described in Algorithm 2.

### 4.4. Theoretical Analysis for SMD

Building on our earlier discussion of how the repeated projection mechanism ensures convex constraint satisfaction, we now present a more detailed theoretical justification of our proposed SMD for MRMP. *The key insight is SMD can provide feasible trajectories for MRMP by introducing the Lagrangian relaxation method. Although the full MRMP problem is nonconvex, we leverage the constraint structure to control violations during each projection step.*

For simplicity, we conduct our analysis on the standard Lagrangian formulation in Eq. (9), as the augmented Lagrangian method mainly enhances the convergence of the original Lagrangian relaxation. Before proceeding, we make the following assumptions commonly satisfied in MRMP formulations.

**Assumption 4.1.** The cost function of relaxed MRMP $\mathcal{L}(\Pi, \nu_a, \nu_o)$ is continuously differentiable and convex over the convex set $\Omega_c$.

**Proposition 4.2** (Convex Feasibility Guarantee). *Let $\Pi$ be the trajectory output by the projection operator $\mathcal{P}_\Omega(\Pi)$ within the diffusion step of SMD. Suppose that Assumption 4.1 holds. Then, the generated trajectory $\Pi$ satisfies*

$$dist(\Pi, \Omega_c) \leq \xi$$

*where $dist(\cdot)$ denotes the distance between the trajectory $\Pi$ and the feasible region $\Omega_c$, and $\xi \geq 0$ can be made arbitrarily small.*

*Remark* 4.3. This proposition provides a theoretical guarantee that our SMD can ensure the feasibility for convex feasible set $\Omega_c$ by introducing Lagrangian relaxation methods. In addition to the convex constraints, SMD handles nonconvex constraints defined by functions $\mathcal{H}_a(\Pi)$ and $\mathcal{H}_o(\Pi)$ via a dual ascent method with the user-defined stopping criterion $\delta_a$ and $\delta_o$. Specifically, SMD ensures the

output of the projection operator $\Pi$:

$$||\mathcal{H}_a(\Pi)|| \leq \delta_a, \quad ||\mathcal{H}_o(\Pi)|| \leq \delta_o,$$

for some tolerances $\delta_a, \delta_o > 0$.

## 5. Experimental Settings

This section describes the evaluation methodology for Multi-Robot Motion Planning algorithms, detailing the benchmark maps, task assignment process, and performance metrics. To ensure a comprehensive evaluation, this paper also introduces a new benchmark instances set that captures a variety of real-world MRMP challenges (released as supplemental material).

**Maps.** We evaluate MRMP algorithms on both randomly generated and real-world-inspired maps.

● **Random maps** include environments with increasing levels of complexity: empty maps test inter-robot collision avoidance in open spaces; basic maps introduce 10 obstacles, requiring navigation within the feasible region that excludes obstacle-occupied areas ; dense maps contain 20 obstacles, significantly restricting movement and increasing planning difficulty. The progression from empty to dense maps allows us to systematically assess the algorithms on spatial constraints and coordination.

● **Practical maps** simulate real-world environments such as warehouses and buildings, where constrained pathways and structured layouts impose additional planning challenges. Corridor maps replicate narrow passages where robots must coordinate movements to pass each other without deadlocks. Shelf maps mirror warehouse storage layouts with tight aisles, requiring precise navigation. Room maps introduce multiple rooms connected by doors, restricting the number of robots that can enter at the same time and requiring careful scheduling to prevent congestion. Compared to random maps, practical maps emphasize not only collision avoidance but also global coordination, as robots must find feasible routes through constrained spaces.

Each scenario includes 25 maps with different obstacle configurations.

**Task assignment.** Start and goal states are assigned differently for random and practical maps. In random maps, they are placed at feasible positions without collisions (see Figure 2), while in practical maps, they are constrained to predefined zones that reflect real-world constraints, such as pickup and drop-off locations in warehouses (see Figure 3). We conduct experiments with 3, 6, and 9 robots, generating 10 test cases for each configuration, except for corridor maps, where we use 2 robots.

**Evaluation metrics.** MRMP algorithms are assessed based on their ability to generate collision-free, efficient, and

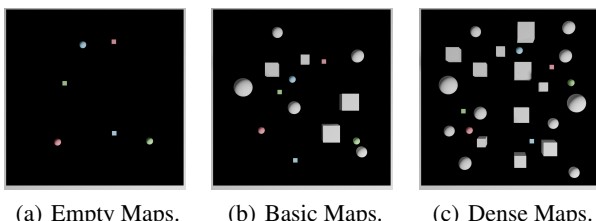

(a) Empty Maps.    (b) Basic Maps.    (c) Dense Maps.

*Figure 2.* Examples of random maps used for MRMP experiments, with increasing complexity. Colorful spheres and plates denote the start and goals of robots. White objects indicate obstacles.

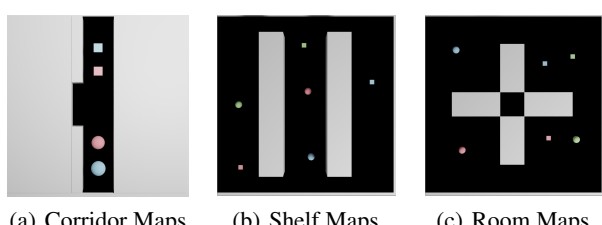

(a) Corridor Maps.    (b) Shelf Maps.    (c) Room Maps.

*Figure 3.* Examples of practical maps used for MRMP experiments show different practical challenges in MRMP.

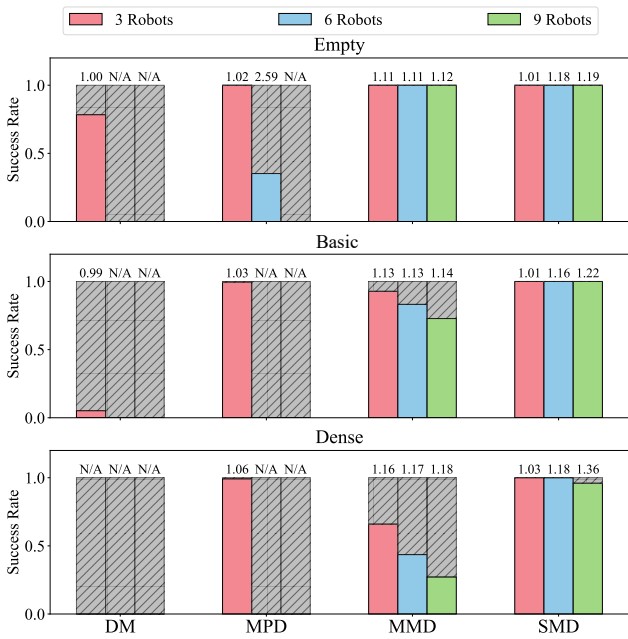

*Figure 4.* Results for each method on random maps with three different numbers of robots. Gray bars represents the failure rate, and values on top of the bars indicate average path length per robot.

smooth trajectories. Success rate measures the percentage of test cases solved without collisions. Path length evaluates the average travel distance per robot, reflecting efficiency. Acceleration quantifies trajectory smoothness, with lower values indicating reduced energy consumption and smoother motion. Collision ratio measures the proportion of robots that experience collisions, providing insight into the robustness of the planning method. These metrics collectively assess feasibility, efficiency, and safety in MRMP.

**Competing Methods.** We compare our proposed SMD against the following learning-based baseline methods:

1. **Diffusion Models (DM)**: We adopt standard diffusion models trained over the feasible trajectories to address MRMP directly (Nichol & Dhariwal, 2021).
2. **Motion Planning Diffusion (MPD)**: The state-of-the-art motion planning diffusion model for single robots (Carvalho et al., 2023), which we extend to handle MRMP for comparison.
3. **Multi-robot Motion Planning Diffusion (MMD)**: A recently introduced solution that combines diffusion models with classical search-based techniques—generating MRMP solutions under collision constraints (Shaoul et al., 2025).

We also compare our proposed approaches with classical algorithms in Appendix C.

## 6. Experiments

In this section, we evaluate the performance of various MRMP algorithms using our proposed benchmark. In most cases, regular projection methods cannot be applied to obtain solutions in MRMP due to the huge computational burden (hence, our omission of Christopher et al. (2024) as a baseline). Thus, for our implementation of SMD, we use the projection with the Lagrangian relaxation method for our experiments, except for the narrow-corridor map. The implementation details can be found in Appendix A. Additional experimental results are presented in Appendix C, including the performance of classical algorithms, additional evaluation metrics, runtime comparisons, sensitivity analysis of projection parameters, and results on other map types.

### 6.1. Performance on Random Maps

We first showcase the methods' performance on random maps. Figures 4 compares the success rate and path length for all methods for settings of increasing complexity and three configurations (3, 6, and 9 robots).

First, notice that the *Standard Diffusion Model (DM)* is unviable for MRMP tasks. Despite reporting a reasonable success rate (78%) for 3 robots in empty maps, when simple obstacles are introduced, the success rate falls dramatically (5%). Additionally, DM fails to produce any feasible trajectory when scaling to six or nine robots, even in empty

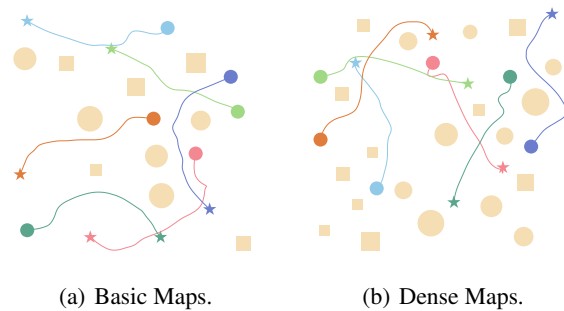

(a) Basic Maps.          (b) Dense Maps.

*Figure 5.* Trajectories generated by SMD on random maps.

maps.

Next, we focus on *Motion Planning Diffusion (MPD)*. While it reports near-perfect success rates for the 3-robots setting (the smallest in our benchmark), this approach is highly ineffective when scaled to additional robots. In empty environments, MPD reports 35% success rates with six robots and it fails to synthesize any feasible trajectory when obstacles are introduced to the map or when scaling to nine robots.

*Multi-robot Motion Planning Diffusion (MMD)* represents the current state-of-the-art method for MRMP. This method outperforms other baselines in its ability to handle an increased number of robots. For instance, it reports non-zero success rates for up to nine robots. However, there remains a steep performance drop with the increasing number of robots. For instance, it reports a success rate of 27% in dense maps for 9 agents. A similar decrease in constraint satisfaction occurs for in other scenarios besides the empty maps.

In contrast, *Simultaneous MRMP Diffusion (SMD)* provides new state-of-the-art results for both constraint satisfaction and path length. Unlike other methods whose success rates rapidly decline as robot and obstacle numbers grow, SMD maintains feasibility in scenarios with increased robot and obstacle counts (e.g., dense maps with 9 robots and 20 obstacles). Specifically, SMD is the only known method that provides feasible solutions for the largest number of robots in complex environments. SMD reports perfect success rates and collision ratios for all tasks except the most complex maps (dense environments) and 9 robots. Even in the most challenging dense maps, where it still provides near-perfect results (96% successful). *This is a* **3.6x** *improvement over the previous state-of-the-art*. Sample trajectories generated by SMD are visualized in Figure 5.

## 6.2. Performance on Practical Maps

Practical maps are designed to evaluate the ability of the methods to generate feasible trajectories for more challeng-

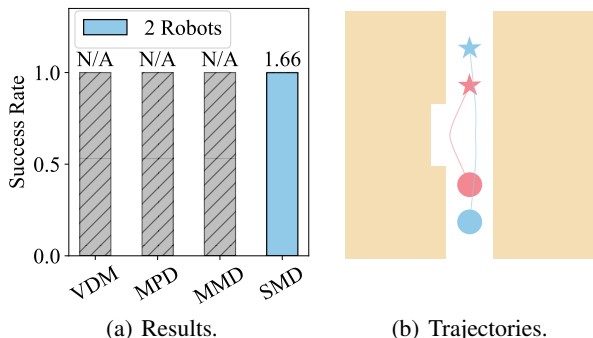

(a) Results.          (b) Trajectories.

*Figure 6.* Results for each method on corridor maps. Gray bars represents the failure rate, and values on top of the bars indicate average path length per robot.

ing yet commonly occurring real-world scenarios.

In *corridor maps*, Figure 6(a) shows that our SMD achieves a perfect success rate, whereas *all other methods fail to report even a single feasible trajectory*. In restrictive environments like this, robots need to swap their positions in a specific area to achieve their goals, as their initial relative positions are opposite to their goals positions, and the narrow corridors do not allow direct position exchange. In this case, the gradient-based guidance used by MPD in the diffusion sampling process is prone to failure, as it cannot globally coordinate different robots to reach their respective goals. Although MMD uses multi-agent path-finding algorithms to account for goal-reasoning, it still relies on gradient-based guidance to generate trajectories. The penalty term employed by this method during gradient computation is thus insufficient to produce feasible solutions in such regions. In contrast, our reformulation of the diffusion sampling process as a constrained optimization problem confers the ability to ensure feasibility in the generated trajectories. As shown in Figure 6(b), the trajectories generated by our SMD allow the two robots to use the only slightly wider area to swap their positions.

We also evaluate on *shelf maps* and *room maps*, both of which contain narrow passages that must be traversed to reach the goals. Figure 7 shows that MPD and MMD achieve a similar success rate (60%) when there are only three robots in shelf maps, and MPD even provides shorter paths. However, it quickly fails to provide feasible solutions as the number of robots increases, even slightly. SMD again achieves a perfect success rate for three robots, and it only experiences a slight decline as the number of robots increases (90%). SMD also reports the shortest paths among all methods. Similarly, *room maps* report SMD achieving a perfect success rate for 3 and 6 robots and above 90% when scaling up to 9 robots. In contrast, MPD and MMD fail to produce feasible trajectories as the number of robots increases. In these maps, SMD advantage over the baselines

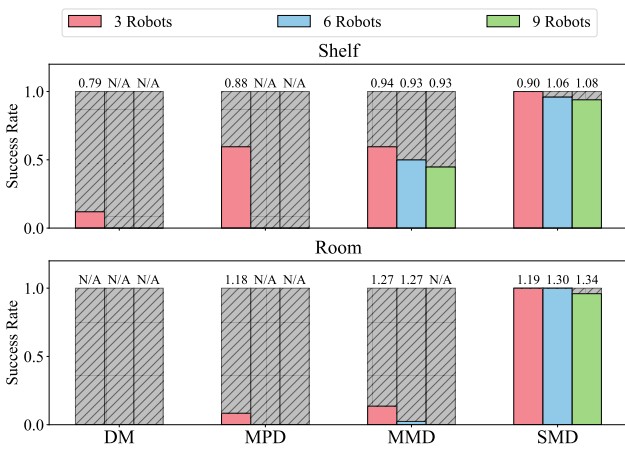

*Figure 7.* Results for each method on practical maps with three different numbers of robots. Gray bars represents the failure rate, and values on top of the bars indicate average path length per robot.

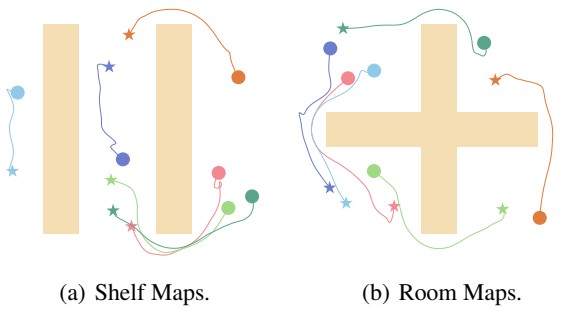

(a) Shelf Maps.       (b) Room Maps.

*Figure 8.* Trajectories generated by SMD on practical maps.

is even more noticeable. These results shows the adaptability of the proposed algorithm to multiple environments and challenging scenarios (such as the narrow passages).

Finally, Figure 8 provides two illustrative cases for shelf maps (a) and room maps (b) where SMD is the only method across those tested able to generate feasible solutions. A common characteristic of these two instances is that multiple robots need to pass through the same narrow passage, a scenario frequently encountered in real-world applications.

## 7. Discussion and Limitations

While SMD provides state-of-the-art results for MRMP tasks, consideration should be given to the appropriate applications of this approach. First, generative models incur inference-time costs that should be considered when applying these models to motion planning. Diffusion models are most advantageous in scenarios where the complexity of the environment and the need for multimodal trajectory generation outweigh the benefits of faster but more

rigid methods. For instance, such architectures are ideal for handling unstructured environments with dense obstacles, dynamic changes, and uncertain sensor inputs, where traditional methods struggle to adapt but may be unnecessary in structured or open spaces.

## 8. Conclusion

This work introduced *Simultaneous MRMP Diffusion (SMD)*, a method that integrates constrained optimization with generative diffusion models to generate collision-free, kinematically feasible trajectories for multi-robot systems. By alternating diffusion sampling with projections onto the feasibility set, SMD eliminates the need for rejection sampling or post-processing. To handle complex nonconvex constraints, the paper incorporates an Augmented Lagrangian Method enabling scalability to challenging multi-robot scenarios where other methods fail. To support rigorous evaluation, a benchmark was also introduced, covering environments that reflect real-world motion planning challenges. Extensive experiments across varying obstacle densities and increasing robot counts demonstrated that SMD achieves significantly higher success rates and better objective values than competing methods. The results highlight its robustness in handling both static obstacles and dynamic interactions, making it a promising approach for real-world applications.

There are several promising directions for future work. Extending SMD to more complex scenarios, such as 3D environments and humanoid locomotion is very promising. Furthermore, the extension of SMD to decentralized settings still remain to be explored.

## Acknowledgments

This research is partially supported by NSF grants 2334936, 2334448, and NSF CAREER Award 2401285. The research at the University of California, Irvine was supported by the NSF grants 2434916, 2346058, 2321786, 2121028, and 1935712, as well as gifts from Amazon Robotics. The authors acknowledge Research Computing at the University of Virginia for providing computational resources that have contributed to the results reported within this paper. The views and conclusions contained in this document are those of the authors and should not be interpreted as representing the official policies, either expressed or implied, of the sponsoring organizations, agencies, or the U.S. government.

## Impact Statement

This paper presents work whose goal is to advance the field of Machine Learning. There are many potential societal consequences of our work, none of which we feel must be specifically highlighted here.

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

# A. Implementation Details

**Software**: The software used for experiments is Rocky Linux release 8.9, Python 3.8, Cuda 11.8, and PyTorch 2.0.0.
**Hardware**: For each of our experiments, we used 1 AMD EPYC 7352 24-Core Processor and 1 NVIDIA RTX A6000 GPU.

## A.1. MRMP Details

In our experiments, the size of each local map was $2 \times 2$ units. The robot radius was set to 0.05 units in all cases, except for the corridor map, where it was 0.1 units. The obstacle sizes in random maps varied between 0.05 and 0.1 units.

To adopt the EECBS algorithm, we discretized the map into a grid with a cell size of 0.1 units, which corresponds to the robot's diameter. If an obstacle was present within a grid cell, the cell was marked as an obstacle. The robot's starting position was determined by the grid cell in which its center was located.

## A.2. Training Details

Our implementation builds upon the official code of Carvalho et al. (2023) and Shaoul et al. (2025), with modifications to accommodate our specific requirements. Since MMD can be trained with single-robot motion planning data, we first trained it accordingly. Then, by running MMD, we obtained the feasible solutions of MRMP it generated and used them as training data. Table 1 shows the hyperparameters used for training the score-based diffusion models in our experiments.

*Table 1.* Hyperparameters for Training in Experiments.

| HyperParameters | Value |
|---|---|
| Diffusion Sampling Step | 25 |
| Learning Rate | 1e-4 |
| Batch Size | 64 |
| Optimizer | Adam |

## A.3. Evaluation Details

We conducted experiments on six types of maps, each with 25 different parameters, such as obstacle size and positions. For all map types except corridor maps, we evaluate three different numbers of robots (3, 6, and 9). In corridor maps, only two robots are tested. For each number of robots, we generate 10 test cases, resulting in 4,000 test instances for each method.

*Table 2.* Summary of MRMP Benchmark Instances

| Map Type | Obstacle Variations | Number of Robots Variations | Start & Goal Variations | Number of Instances |
|---|---|---|---|---|
| Empty Maps | 25 | 3 | 10 | 750 |
| Basic Maps | 25 | 3 | 10 | 750 |
| Dense Maps | 25 | 3 | 10 | 750 |
| Corridor Maps | 25 | 1 | 10 | 250 |
| Shelf Maps | 25 | 3 | 10 | 750 |
| Room Maps | 25 | 3 | 10 | 750 |
| Total | | | | 4000 |

For each generated path, we first check for collisions and compute the success rate of each method. For collision-free cases, we analyze path length and acceleration. For cases with collisions, we report the collision ratio. If a method fails to generate a feasible solution, its path length and acceleration are excluded from the statistical analysis.

## B. Missing Proofs

### B.1. Proof of Proposition 4.2

*Proof.* By Assumption 4.1, the projection operator $\mathcal{P}_\Omega(\boldsymbol{x})$ performs the following optimization problem at each iteration:

$$\boldsymbol{\Pi} = \arg \min_{\boldsymbol{\Pi} \in \Omega_c} \mathcal{L}(\boldsymbol{\Pi}, \boldsymbol{\nu}_a, \boldsymbol{\nu}_o). \tag{13}$$

Since $\Omega_c$ is convex and $\mathcal{L}(\boldsymbol{\Pi}, \boldsymbol{\nu}_a^k, \boldsymbol{\nu}_o^k)$ is convex and continuously differentiable over $\Omega_c$, the optimization problem has a unique global minimum.

We define a single update step for the diffusion sampling process as:

$$\mathcal{U}(\boldsymbol{\Pi}_t^i) = \boldsymbol{\Pi}_t^i + \gamma_t \mathbf{s}_\theta(\boldsymbol{\Pi}_t^i, t) + \sqrt{2\gamma_t}\mathbf{z}. \tag{14}$$

For each time step $t$, there exists a minimum iteration index $i = \bar{I}$ such that:

$$\exists \bar{I} \text{ s.t. } \left\| (\boldsymbol{\Pi}_t^{\bar{I}} + \gamma_t \nabla_{\boldsymbol{\Pi}_t^i} \log q(\boldsymbol{\Pi}_t | \boldsymbol{\Pi}_0)) \right\|_2 \leq \|F_t\|_2 \tag{15}$$

where $F_t$ is the closest point to the global optimum that can be reached via a single gradient step from any point in $\Omega_c$.

Let $Error$ be the distance between $\boldsymbol{\Pi}_t^i$ and its nearest feasible point. Using Theorem 5.2 in (Christopher et al., 2024), for any $i \geq I$, we have:

$$\mathbb{E}\left[Error(\mathcal{U}(\mathcal{P}_{\Omega_c}(\boldsymbol{\Pi}_t^i)), \Omega_c)\right] \leq \xi \leq \mathbb{E}\left[Error(\mathcal{U}(\boldsymbol{\Pi}_t^i), \Omega_c)\right]. \tag{16}$$

where $\xi \geq 0$ is arbitrarily small.

Therefore, the expected distance between the final generated trajectory and the feasible set $\Omega_c$ is bounded above by $\xi$, which means our SMD method yields a strictly smaller expected constraint violation and provides a feasibility guarantee for the convex constraint set $\Omega_c$.

$\square$

## C. Additional Experimental Results

Our manuscript primarily evaluates learning-based algorithms based on their performance in terms of success rate and path length (Figure 4, Figure 6(a), and Figure 7). While these metrics are sufficient to demonstrate SMD's ability to consistently outperform its competitors, two additional aspects warrant further investigation: (1) comparison with traditional methods and (2) solution smoothness—measured by acceleration (computed as the absolute difference in velocity between consecutive time steps)—and collision ratio, which quantifies the proportion of robots that collide when no feasible solution is found. Specifically, we discretize the environment to adopt a powerful near-optimal multi-agent path finding algorithm, Explicit Estimation Conflict-Based Search (EECBS) (Li et al., 2021).

As shown in Table 3 and Table 4, our proposed SMD consistently achieves an impressive success rate across all tested scenarios. In contrast, traditional approaches such as EECBS also maintain a high success rate but struggle with path length, which is approximately 20% longer than that of our method. This is caused by they operate within a predefined grid structure. Notably, our SMD achieves near-zero collision rates, outperforming other learning-based approaches, which frequently fail to avoid collisions in complex scenarios. The computational costs are reported in Table 5. Notably, more challenging tasks typically take longer to solve, thus increasing the average running time.

In addition, Table 6 shows evaluation results on practical maps introduced by (Shaoul et al., 2025). SMD still maintains a zero collision rate. These results further confirm the practical applicability and reliability of our approach.

In Figures 9 and 10, we present a sensitivity analysis of the scaling factor $\alpha$ used in our ALM-based projection. The results suggest that careful tuning of $\zeta$ is not necessary; using a moderate value such as 1.05 already achieves good convergence performance.

*Table 3.* Additional results for classical algorithms and learning-based algorithms in random maps. **S** is the success rate, **L** denotes the average path length per robot, **A** is the average acceleration, and **C** is the collision ratio (see Section 5). We omit acceleration and collision information for multi-agent path finding methods, as they assume constant velocities and typically do not return infeasible solutions.

| Map | Robots | Metric | EECBS | DM | MPD | MMD | Ours |
|---|---|---|---|---|---|---|---|
| Empty Maps | 3 | S ↑ | 1 | 0.7840 | 1 | 1 | 1 |
| | | L ↓ | 1.2591 | 0.9997 | 1.0180 | 1.1109 | 1.0125 |
| | | A ↓ | – | 0.0030 | 0.0022 | 0.0028 | 0.0010 |
| | | C ↓ | – | 0.1493 | 0 | 0 | 0 |
| | 6 | S ↑ | 1 | 0 | 0.3520 | 1 | 1 |
| | | L ↓ | 1.2607 | – | 2.5886 | 1.1139 | 1.1764 |
| | | A ↓ | – | – | 0.0062 | 0.0029 | 0.0034 |
| | | C ↓ | – | 1 | 0.4167 | 0 | 0 |
| | 9 | S ↑ | 1 | 0 | 0 | 1 | 1 |
| | | L ↓ | 1.2684 | – | – | 1.1168 | 1.1945 |
| | | A ↓ | – | – | – | 0.0031 | 0.0046 |
| | | C ↓ | – | 1 | 0.9929 | 0 | 0 |
| Basic Maps | 3 | S ↑ | 1 | 0.0520 | 0.9960 | 0.9280 | 1 |
| | | L ↓ | 1.3236 | 0.9910 | 1.0310 | 1.1315 | 1.0091 |
| | | A ↓ | – | 0.0031 | 0.0012 | 0.0032 | 0.0040 |
| | | C ↓ | – | 0.6133 | 0.0013 | 0.0467 | 0 |
| | 6 | S ↑ | 1 | 0 | 0 | 0.8320 | 1 |
| | | L ↓ | 1.3228 | – | – | 1.1319 | 1.1560 |
| | | A ↓ | – | – | – | 0.0033 | 0.0120 |
| | | C ↓ | – | 1 | 0.9847 | 0.0813 | 0 |
| | 9 | S ↑ | 1 | 0 | 0 | 0.7280 | 1 |
| | | L ↓ | 1.3244 | – | – | 1.1375 | 1.2212 |
| | | A ↓ | – | – | – | 0.0035 | 0.0048 |
| | | C ↓ | – | 1 | 1 | 0.1093 | 0 |
| Dense Maps | 3 | S ↑ | 0.968 | 0 | 0.9920 | 0.6600 | 1 |
| | | L ↓ | 1.4945 | – | 1.0568 | 1.1638 | 1.0337 |
| | | A ↓ | – | – | 0.0014 | 0.0037 | 0.0025 |
| | | C ↓ | – | 0.9093 | 0.0040 | 0.1813 | 0 |
| | 6 | S ↑ | 0.944 | 0 | 0 | 0.4360 | 1 |
| | | L ↓ | 1.5135 | – | – | 1.1693 | 1.1841 |
| | | A ↓ | – | – | – | 0.0038 | 0.0064 |
| | | C ↓ | – | 1 | 1 | 0.2687 | 0 |
| | 9 | S ↑ | 0.932 | 0 | 0 | 0.2720 | 0.9600 |
| | | L ↓ | 1.5349 | – | – | 1.1793 | 1.3556 |
| | | A ↓ | – | – | – | 0.0041 | 0.0052 |
| | | C ↓ | – | 1 | 1 | 0.3707 | 0.0044 |

*Table 4.* Additional results for classical algorithms and learning-based algorithms in practical maps.

| Map | Robots | Metric | EECBS | DM | MPD | MMD | Ours |
|---|---|---|---|---|---|---|---|
| Corridor Maps | 2 | S ↑ | 1 | 0 | 0 | 0 | 1 |
| | | L ↓ | 1.4208 | – | – | – | 1.6619 |
| | | A ↓ | – | – | – | – | 0.0012 |
| | | C ↓ | – | 1 | 1 | 1 | 0 |
| Shelf Maps | 3 | S ↑ | 1 | 0.1200 | 0.5960 | 0.5960 | 1 |
| | | L ↓ | 1.1697 | 0.7884 | 0.8825 | 0.9371 | 0.9017 |
| | | A ↓ | – | 0.0030 | 0.0081 | 0.0030 | 0.0060 |
| | | C ↓ | – | 0.5027 | 0.2493 | 0.3133 | 0 |
| | 6 | S ↑ | 1 | 0 | 0 | 0.5000 | 0.9560 |
| | | L ↓ | 1.1493 | – | – | 0.9317 | 1.0632 |
| | | A ↓ | – | – | – | 0.0030 | 0.0018 |
| | | C ↓ | – | 1 | 0.9960 | 0.3740 | 0.01 |
| | 9 | S ↑ | 1 | 0 | 0 | 0.4480 | 0.9320 |
| | | L ↓ | 1.1427 | – | – | 0.9331 | 1.0838 |
| | | C ↓ | – | – | – | 0.0032 | 0.0068 |
| | | A ↓ | – | 1 | 1 | 0.4218 | 0.0111 |
| Room Maps | 3 | S ↑ | 1 | 0 | 0.0840 | 0.1360 | 1 |
| | | L ↓ | 1.6956 | – | 1.1771 | 1.2697 | 1.1917 |
| | | A ↓ | 0 | – | 0.0066 | 0.0043 | 0.0030 |
| | | C ↓ | – | 1 | 0.6520 | 0.6867 | 0 |
| | 6 | S ↑ | 0.9960 | 0 | 0 | 0.0240 | 1 |
| | | L ↓ | 1.6778 | – | – | 1.2660 | 1.3019 |
| | | A ↓ | – | – | – | 0.0032 | 0.0014 |
| | | C ↓ | – | 1 | 1 | 0.8033 | 0 |
| | 9 | S ↑ | 0.9920 | 0 | 0 | 0 | 0.9600 |
| | | L ↓ | 1.6620 | – | – | – | 1.3374 |
| | | A ↓ | – | – | – | – | 0.0033 |
| | | C ↓ | – | 1 | 1 | 0.8849 | 0.0044 |

*Table 5.* Running time in seconds with success rates (shown in parentheses and expressed as percentages) for all learning-based methods across all maps. We consider only the time consumed to generate feasible trajectories for MMD, as the running time for failed trajectories in MMD is determined by a predefined time limit, which could be meaningless if we calculate the running time using time limit (e.g., 3000s).

| Map | Robots | DM | MPD | MMD | Ours |
|---|---|---|---|---|---|
| Empty Maps | 3 | 4.2(78.4) | 14.1(**100**) | 37.6(**100**) | 29.5(**100**) |
| | 6 | 3.7(0) | 11.7(35.2) | 72.1(**100**) | 103.8(**100**) |
| | 9 | 3.9(0) | 9.9(0) | 108.5(**100**) | 204.7(**100**) |
| Basic Maps | 3 | 4.1(5.2) | 14.3(99.6) | 39.6(92.8) | 75.1(**100**) |
| | 6 | 4.1(0) | 9.9(0) | 81.9(83.2) | 235.2(**100**) |
| | 9 | 4.1(0) | 10.2(0) | 123.1(72.8) | 481.6(**100**) |
| Dense Maps | 3 | 3.8(0) | 14.5(99.2) | 46.3(66.0) | 73.7(**100**) |
| | 6 | 4.3(0) | 10.1(0) | 105.3(43.6) | 255.1(**100**) |
| | 9 | 4.1(0) | 10.3(0) | 192.4(27.2) | 551.1(**96.0**) |
| Shelf Maps | 3 | 4.1(12.0) | 9.3(59.6) | 59.2(59.6) | 91.3(**100**) |
| | 6 | 4.1(0) | 7.6(0) | 128.4(50.0) | 283.2(**95.6**) |
| | 9 | 4.1(0) | 8.2(0) | 200.2(44.8) | 585.9(**93.2**) |
| Room Maps | 3 | 4.1(0) | 7.9(8.4) | 20.5(13.6) | 74.6(**100**) |
| | 6 | 4.1(0) | 8.8(0) | 41.2(2.4) | 158.7(**100**) |
| | 9 | 4.1(0) | 8.5(0) | N/A(0) | 255.7(**96.0**) |

*Table 6.* Additional results for MMD and SMD in maps used in (Shaoul et al., 2025).

| Map | Robots | Metric | MMD | Ours |
|---|---|---|---|---|
| | | S ↑ | 1 | 1 |
| | 3 | L ↓ | 1.1638 | 1.1391 |
| | | A ↓ | 0.0026 | 0.0030 |
| | | C ↓ | 0 | 0 |
| | | S ↑ | 1 | 1 |
| Highways Maps | 6 | L ↓ | 1.1669 | 1.1775 |
| | | A ↓ | 0.0028 | 0.0022 |
| | | C ↓ | 0 | 0 |
| | | S ↑ | 1 | 1 |
| | 9 | L ↓ | 1.1792 | 1.1802 |
| | | A ↓ | 0.0028 | 0.0031 |
| | | C ↓ | 0 | 0 |
| | | S ↑ | 1 | 1 |
| | 3 | L ↓ | 1.1529 | 1.1499 |
| | | A ↓ | 0.0030 | 0.0029 |
| | | C ↓ | 0 | 0 |
| | | S ↑ | 1 | 1 |
| Conveyor Maps | 6 | L ↓ | 1.1726 | 1.1806 |
| | | A ↓ | 0.0027 | 0.0044 |
| | | C ↓ | 0 | 0 |
| | | S ↑ | 1 | 1 |
| | 9 | L ↓ | 1.1916 | 1.2003 |
| | | A ↓ | 0.0031 | 0.0067 |
| | | C ↓ | 0 | 0 |
| | | S ↑ | 1 | 1 |
| | 3 | L ↓ | 1.1417 | 1.0337 |
| | | A ↓ | 0.0024 | 0.0051 |
| | | C ↓ | 0 | 0 |
| | | S ↑ | 1 | 1 |
| Drop-Region Maps | 6 | L ↓ | 1.1604 | 1.1841 |
| | | A ↓ | 0.0028 | 0.0039 |
| | | C ↓ | 0 | 0 |
| | | S ↑ | 1 | 1 |
| | 9 | L ↓ | 1.1719 | 1.3556 |
| | | A ↓ | 0.0027 | 0.0075 |
| | | C ↓ | 0 | 0 |

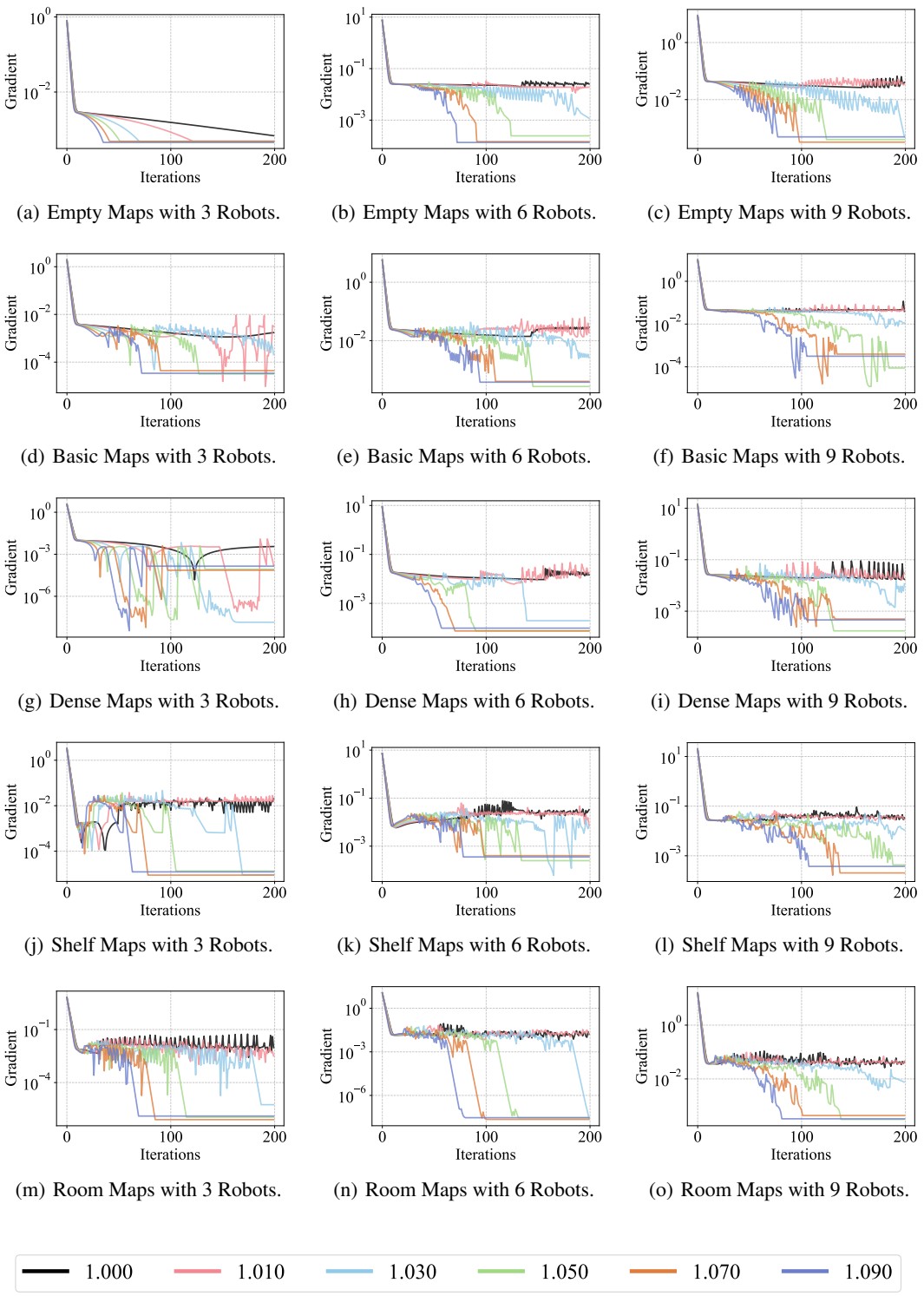

*Figure 9.* Sensitivity analysis of the scaling factor $\zeta$ on gradient convergence for inter-agent collision avoidance constraints in ALM-based projection, evaluated for each map with different numbers of robots. The scaling factor $\zeta$ determines how the coefficient of the augmented term is multiplied at each iteration. We test five increasing values of $\zeta$ from 1.01 to 1.09, as well as a constant value of 1.00 as the ablation study.

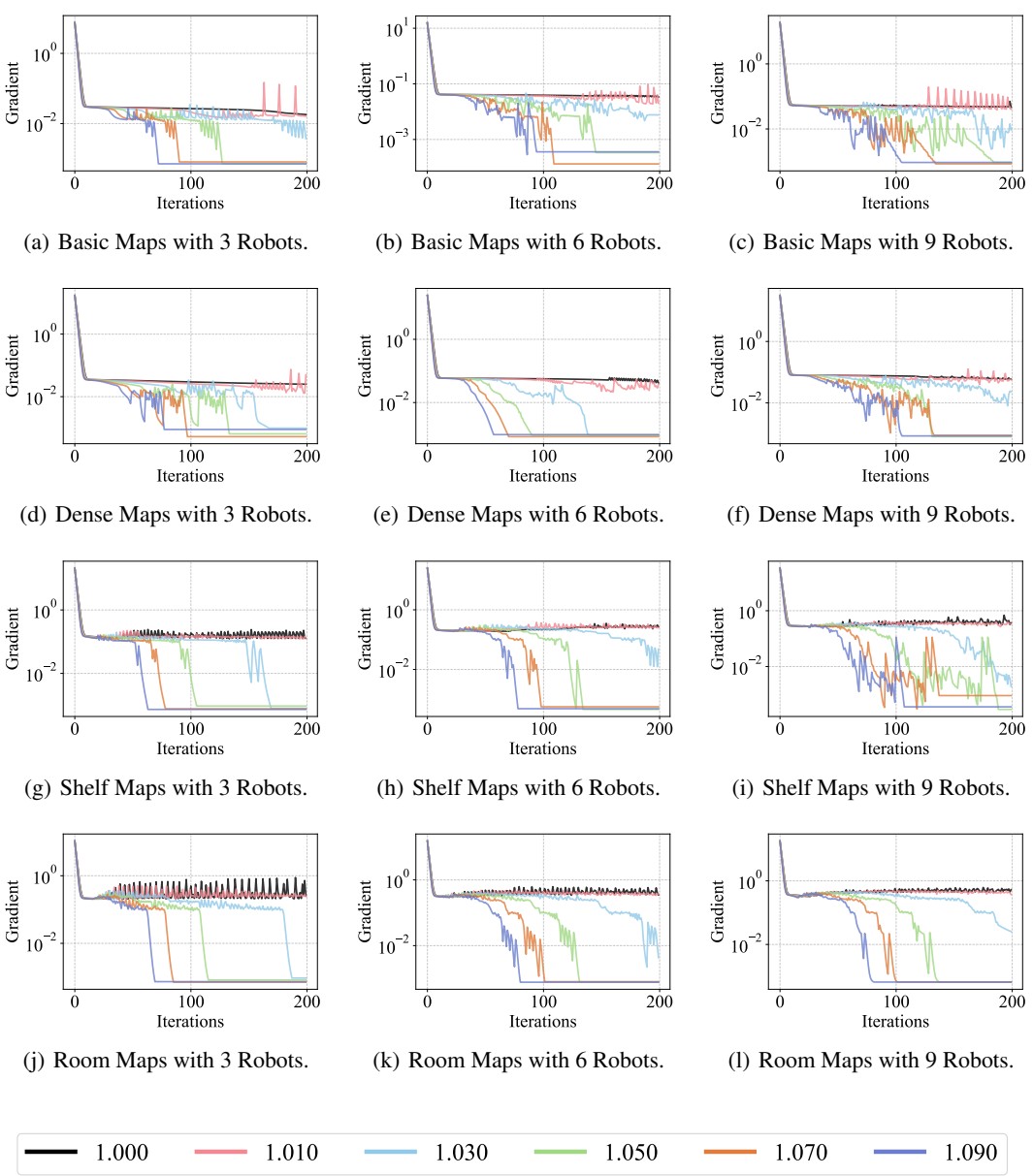

(a) Basic Maps with 3 Robots.  (b) Basic Maps with 6 Robots.  (c) Basic Maps with 9 Robots.

(d) Dense Maps with 3 Robots.  (e) Dense Maps with 6 Robots.  (f) Dense Maps with 9 Robots.

(g) Shelf Maps with 3 Robots.  (h) Shelf Maps with 6 Robots.  (i) Shelf Maps with 9 Robots.

(j) Room Maps with 3 Robots.  (k) Room Maps with 6 Robots.  (l) Room Maps with 9 Robots.

*Figure 10.* Sensitivity analysis of the scaling factor on gradient convergence for obstacle collision avoidance constraints in ALM-based projection, which leads to similar conclusions with inter-agent collision avoidance constraints.

