# OpenReview forum: "Simultaneous Multi-Robot Motion Planning with Projected Diffusion Models"
_ICML.cc/2025/Conference — ICML 2025 poster_

### Official Review · Reviewer_LLNt · 2025-03-05

**Overall Recommendation:** 3

**Summary:**

The paper proposes Simultaneous Multi-Robot Motion Planning with Projected Diffusion Models (SMD), a novel method for multi-robot motion planning (MRMP) that integrates constrained optimization into the sampling process of diffusion models. Although diffusion models have demonstrated promising capabilities in generating diverse and smooth robot trajectories, they often fail to satisfy essential constraints, including collision avoidance and kinematic feasibility. To address this, the authors introduce a constrained optimization approach embedded within the diffusion model's sampling process using an augmented Lagrangian method. This formulation efficiently projects generated trajectories into feasible spaces, ensuring compliance with both collision avoidance and kinematic constraints. Additionally, the authors provide a comprehensive new benchmark dataset to evaluate multi-robot motion planners across scenarios with varying obstacle densities and spatial complexities. Empirical results show that SMD significantly outperforms other baselines.

**Claims And Evidence:**

Yes

**Essential References Not Discussed:**

No

**Experimental Designs Or Analyses:**

I find the experimental designs reasonable and sufficiently aligned with the paper’s objectives.

**Methods And Evaluation Criteria:**

Yes

**Other Comments Or Suggestions:**

While the core contribution is solid, additional details about computational overhead—particularly in comparison to MMD—would strengthen the paper.

Providing at least one reproducible example or making the code available would greatly help readers verify the claims and understand how easily the proposed SMD method can be integrated into existing robot planning systems or simulation environments.

**Other Strengths And Weaknesses:**

Strengths:

The paper is well-written, and the proposed method is described clearly.

The combination of diffusion models with constrained optimization to ensure feasibility seems both novel and promising.

The comparisons against a strong baselines (including MMD) in the multi-robot motion planning setting help validate the method’s effectiveness.

Weaknesses:

It remains unclear how sensitive the method’s performance is to certain hyperparameters introduced by constrained optimization, and how difficult they are to tune in practice.

The paper lacks discussions about reproducibility and practical integration.

**Questions For Authors:**

How does SMD’s inference time scale with the number of robots (e.g., up to 100)? Do you have experimental evidence indicating whether it remains tractable for large-scale multi-robot scenarios?

Could you provide some results / explaintions on how senstive the performance of the proposed approach is to the additional hyperparameters introduced and how hard it is tune them?

In the early diffusion steps, when the trajectory is still highly noisy, wouldn't strongly enforcing constraints potentially lead to convergence towards unrealistic trajectories?

While the proposed benchmark is a valuable contribution, I have some concern that it might be biased toward your method’s strengths. Have you tested SMD on other existing benchmarks (e.g., those used in MMD) to confirm consistent performance improvements? Additionally, do you expect that SMD can handle more complex robotic systems (e.g., humanoid locomotion or higher-dimensional configuration spaces), rather than just 2D disk navigation?

**Relation To Broader Scientific Literature:**

This paper builds on prior score-based diffusion approaches and extends them by incorporating an augmented Lagrangian framework to handle multi-robot constraints, bridging a gap in the literature where most methods either rely on gradient-based cost penalties or post-hoc filtering for feasibility.

**Theoretical Claims:**

No theoretical section is presented in the paper.

---

> ### Author Rebuttal · Authors · 2025-03-31
>
> Thank you for the helpful review, in particular the acknowledgement of the **strong empirical performance** of our proposed SMD and the contribution of **a comprehensive benchmark** for MRMP. We provide below our answers to your insightful questions.
>
> - **Q1: SMD’s inference time with more robots (e.g., up to 100) & experiments with large-scale scenarios?**
>
> It would be amazing to succeed in such scenarios. However, these usually require much-simplified problems, including using a discretization of the underlying map in which multi-agent path finding (MAPF) formulation arises, and assumptions on fixed velocity and discretized environment
>
> Please note that our proposed method scales to the **largest number of robots** in complex scenarios tested so far in this realistic setting, thus setting a new SOTA standard. Additionally, our newely proposed benchmark provides the most comprehensive robot scaling of any existing baselines. We are hopeful that our benchmark will be a useful contribution to the community.
>
> We have also included additional inference times. Please refer to our response to Q3 from reviewer ZshS. Many thanks!
>
>
> - **Q2: Senstivity analysis and tuning for the additional hyperparameters introduced**
>
> Thank you for bringing up this point. During rebuttal, we have conducted an extensive senstivity analysis. We ask you to refer to our response to A3 from reviewer ZshS for more details.
> The empirical observations strongly suggest that our method, being able to integrate constraints directly in the sampling phase of diffusion models, is very robust across a range of hyperparameters, thus reducing the need for meticulous tuning. This is a strength that we plan to discuss in the final version of the paper.
>
> - **Q3: Concerns on generating unrealistic trajectories**
>
> While the reviewer suggests that projecting early on in the diffusion process may "lead to convergence towards unrealistic trajectories," in fact, the opposite is consistently observed. First, consider a case where a single post-processing projection is used to correct the final sample $x_0$. If $x_0$ is far from the constraint set, the projection will alter the trajectories significantly, leading to a lower degree of realism. This is well documented by prior literature [1,2] and has motivated constraint imposition earlier on in the sampling process. By projecting earlier in the diffusion process, the following denoising steps will restore any reaslism that is lost by projecting. Instead, we converge to a feasible subdistribution of the training data.
>
> Practically, the projection starting point could be tuned to balance the tradeoff between computational overhead and generation realism [2], but we defer this type of analysis to future research.
>
> - **Q4: Performance evaluation on SMD's Benchmark & complex robotic systems**
>
> Indeed, maps used in SMD's benchmarks contain fewer obstacles, under which both methods (ours and prior baselines) easily succeed. On more challenging maps, such as dense maps, our method demonstrates clear advantages.
> The results on the original MMD benchmark can be found here:
> https://drive.google.com/file/d/1fOCDaNoU6AP9-f5AvMuurvYSHsogwuNR/view?usp=drive_link
> We would be happy to include them in the final version of the paper if the reviewer suggests so.
>
> Regarding more complex scenarios, while this paper focuses on motion planning in environments with static obstacles, we agree that extending SMD to handle more complex systems (e.g., humanoid locomotion) is an exciting direction for future work. Indeed, our group is active in these directions. One possible extension is to incorporate human trajectory prediction into the projection mechanism, or to adopt a MPC framework to adapt to dynamic agents.
>
> **Reviewer’s Additional Points**:
>
> > **A1**:Reproducibility and practical integration
>
> We agree and intend to release our code following the review cycle. It had previously been our intent to provide an anonymized repository with code examples for the rebuttal, but due to ICML's updated policy, we are only allowed to include figures, tables and explanations related to them in linked material. Our lab has a long tradition of sharing codes and associated github pages with execution instructions and tutorials.
>
> As far as practical integration, our method is training-free and thus can be directly embedded into existing methods to improve performance. **This is also what allows us to generalize to unseen maps!**. The method is easy to use and requires minimal modification to existing pipelines.
>
>
> ---
>
> Thank you for your time assessing our work! We appreciate your suggestions and will add the additional results that your review motivated to our subsequent draft. We would be grateful if you could consider increasing your score to reflect this. Many thanks!
>
> [1] Christopher, Jacob K., et al. "Constrained synthesis with projected diffusion models."
>
> [2] Yuan, Ye, et al. "Physdiff: Physics-guided human motion diffusion model."

---

### Official Review · Reviewer_ZshS · 2025-03-11

**Overall Recommendation:** 3

**Summary:**

The paper tackles constraint enforcement for trajectory generation with diffusion models in the context of multi-agent motion planning. Instead of encoding constraints as auxiliary energy terms, the paper proposes to project intermediate generations on to the collision-free manifold. The projection operator is tackled by dual ascent method so the formulation can be more tractable than solving the original constrained optimization problem. The method is shown to outperform standard diffusion model and variants with planning in-the-loop, in particular in complex problems with more agents and map layout entailing coordination.

**Claims And Evidence:**

The central claim lies in using projection to enforce collision avoidance constraints is more amenable to generate solutions for complex multi-agent planning problems. This is validated by the results on dense and real-world maps in Figure 3.

**Essential References Not Discussed:**

I am not aware of other essential references to be discussed.

**Experimental Designs Or Analyses:**

The experiment designs look sound to verify the acclaimed challenge of applying diffusion model to multi-agent motion planning and the effectiveness of projection based constraint enforcement. The analysis focuses on success rate of solving the problem and shows clear benefits of the proposed method in handling more complex setups.

**Methods And Evaluation Criteria:**

The method follows literature-based formulation and dual ascent method for the effectiveness on solving constrained optimization. The reasoning looks solid although the insights about why the idea can bring about such effectiveness could be better addressed. The main issue on the method is that it seems heavily relying on the quadratic form of constraints that can only capture inter-distance between sphere-shaped robots. It is unclear whether other types of constraints can also be handled by the framework.

The evaluation is made on collision-free motion planning problems on maps with different levels of complexity and number of agents. The criteria on success rate makes sense.

**Other Comments Or Suggestions:**

Line 164: "this allows it to from" missing a verb here?

**Other Strengths And Weaknesses:**

The paper clarity can be improved as the method in beginning reads like a blend of sampling and gradient-based optimization. The details on training implementation is placed in the appendix so it was only clear to me at the end that the paper is about regressing solver's results. The clarity is also very demanding for the implementation of training. Algorithm 2 seems requires to differentiate though an optimization process and it is not clear from the paper whether the gradients are attained through unrolling the optimization process or adjoint methods.

The method seems to have some design space to explore, e.g. coefficients of constraint residual norms. An ablation study can better demonstrate the robustness of the method on the parameter choice.

**Questions For Authors:**

1. How can the framework be used to handle general multi-agent planning constraints beyond collision avoidance?

2. The problem seems not to cover anonymous agent planning. Can it be used on this category of problems?

3. What is the computational cost for the inference of diffusion model with embedded projection. Can it afford some replanning to close the loop?

**Relation To Broader Scientific Literature:**

The idea of leveraging alternating gradient descent to enforce constraints in denoising process seems applicable to broader works on embedding differential optimization in the learning loop. However, it is unclear whether the relaxation can work for non convex constraints other than a quadratic form which works for capturing the inter-agent collision avoidance but might be limited for general constraint enforcement.

**Theoretical Claims:**

There is no theoretical result presented in the paper.

---

> ### Author Rebuttal · Authors · 2025-03-31
>
> Thank you for your valuable feedback including acknowledging the **superior performance of our SMD in complex scenarios**. We provide below our answers to your valuable questions:
>
> **Questions For Authors:**
>
> - **Q1: How can the framework be used to handle general multi-agent planning constraints beyond collision avoidance?**
>
> We agree that quadratic form of constraints used in our work usually captures inter-distance between sphere-shaped robots. Such a formulation is also **widely adopted by existing methods** (e.g., the previous SOTA, MPD and MMD used in our experiments) and can be found in practical environments [1]. If extended to other robot shapes like rectangles, our framework still works but with more complex constraints. A simple alternative is to use the minimum bounding sphere, or multiple spheres to approximate the irregular shapes.
>
> We would like to clarify we do not *only* consider inter-distance constraints, but also **ensure the kinematical constraints (e.g., velocity limits)** in our paper. Additionally, SMD can handle other constraints such as acceleration and smoothness constraints easily by adding these to the constraint set. This could be a meaningful future direction. We will further elaborate in the final version of the paper.
>
> - **Q2: The problem seems not to cover anonymous agent planning. Can it be used on this category of problems?**
>
> Provided that anonymous agent planning is, in fact, a less constrained problem than MRMP, our method can be employed without changes if we preassign goals to agents. For better performance, we can modify the constraints governing the initial and final states of the agents.
>
> - **Q3: What is the computational cost for the inference of diffusion model with embedded projection. Can it afford some replanning to close the loop?**
>
> With a tolerable increase in runtime, our method achieved significant performance improvement over the previously SOTA methods! The detailed comparison is available at this link: https://drive.google.com/file/d/10AnaJkw5Sva8qdNtXbexmNst80pKKjnF/view?usp=drive_link. Our running time can be further optimized using multiple techniques, such as parallelism and learning to optimize methods.
>
> Our method can be easily adapted to support replanning. In fact, this is our current focus. To do so, we adjust the input to incorporate the existing trajectories and perform local updates according to the changed conditions. This approach also significantly reduces computational overhead!
>
> **Reviewer’s Additional Points**:
>
> > **A1**: The reasoning behind the effectiveness of our method, and whether it generalizes beyond quadratic inter-distance constraints
>
> For general constraints, please refer to our response to Q1 for details.
>
> For theoretical and empirical analysis for the effectiveness of our SMD, please refer to our response to Q1 from Reviewer 7Hco.
>
> > **A2**: Clarifying our sampling process
>
> We're happy to clarify this, as it seems there are a few key misunderstandings of our work. First, there seems to be a misunderstanding regarding our approach, which is not merely a "blend of sampling and gradient-based optimization".
>
> The gradients are computed based on the constraint residuals iteratively, which is a common practice in the *dual ascent method* in constrained optimization. Specifically, our approach leverages an optimization solver to project the infeasible trajectories — generated during the diffusion sampling process (from noise to trajectory) — onto the feasible region. This allows us to achieve SOTA performance.
>
> Second, our paper's contribution is centered on extending the *sampling process* and is consequentially applied at *inference time only*. We defer details about training to the appendix because this is not where our work is centered. This aspect however, is important: **it is what allow us to generalize beyond training distribution and handle maps, even if they were never seen before**.
>
> We will be happy to include the training details in the main text and provide a more detailed description of Algorithm 2 in the final version.
>
> > **A3**: Coefficients of constraint residual norms
>
> During rebuttal, we evaluated the convergence performance with 6 different coefficients for the constraint residual norms. The results are available at: https://drive.google.com/file/d/1jjlOs9GEYd2qE4X6OLd4wjEoR3PVkDwo/view?usp=drive_link
>
> Importantly, there is no change w.r.t. the paper's conclusions.
>
> ---
>
> We appreciate your efforts and insightful feedback! We believe our response has addressed each of your questions but would be happy to provide more details if requested. As the points covered in your review are primarily requests for clarifiying, we would be grateful if you could consider increasing your score to reflect this. Many thanks!
>
> [1] Clearpath Robotics. *TurtleBot 4*. https://clearpathrobotics.com/turtlebot-4/
> [2] Rockafellar, R. T. Augmented Lagrange multiplier functions and duality in nonconvex programming.

---

### Official Review · Reviewer_9rTn · 2025-03-12

**Overall Recommendation:** 3

**Summary:**

This paper proposes a new method for tackling the constraint satisfaction challenge in Multi-Robot Motion Planning (MRMP). The paper highlights challenges in existing methods, such as learning-based approaches, that struggle with obeying hard constraints. The proposed method, SMD, addresses these issues by incorporating constrained optimization into diffusion models. To handle nonconvex constraints efficiently, SMD employs an augmented Lagrangian method. This paper also proposes a MRMP benchmark with varied scenarios. SMD outperforms state-of-the-art MRMP methods in complex multi-robot settings.

**Claims And Evidence:**

1. The submission claims that SMD maintains feasibility as robot/obstacle density increases, however, MRMP complexity grows combinatorially with robot count.
2. The main contribution—integrating constrained optimization into diffusion via projections—appears closely aligned with Christopher et al. (2024). The primary adaptation here is applying this framework to MRMP and replacing the solver with an augmented Lagrangian method.

**Essential References Not Discussed:**

No

**Experimental Designs Or Analyses:**

Yes

**Methods And Evaluation Criteria:**

Yes

**Other Comments Or Suggestions:**

It would be helpful if the authors could clarify any additional novelty/contributions.

**Other Strengths And Weaknesses:**

Strengths:
1. The paper demonstrates the effectiveness of the proposed method.
2. The paper introduces a comprehensive benchmark tailored to MRMP.

Weaknesses:
The main contributions closely follow Christopher et al. (2024), with the main adaptations being its application to MRMP and the use of an augmented Lagrangian solver.

**Questions For Authors:**

Differentiation from Christopher et al. (2024)

**Relation To Broader Scientific Literature:**

Generative diffusion models, Multi-Robot Motion Planning

**Theoretical Claims:**

No proofs.

---

> ### Author Rebuttal · Authors · 2025-03-31
>
> We thank Reviewer 9rTn for the insightful feedback including acknowledging the **effectiveness** of our proposed SMD and **comprehensive benchmark** for evaluating MRMP. We provide below our answers to your valuable questions:
>
> - **Differentiation from Christopher et al. (2024)**
>
> Indeed, our work draws inspiration from Christopher et al. (2024), but our contribution is far from being an application of the such framework. It is a significant extension specifically tailored to MRMP.
> Our first key contribution is a diffusion-based method explicitly designed to overcome the challenge of ensuring trajectory feasibility in MRMP. Our SMD can consistently **generate feasible trajectories in complex MRMP scenarios where existing state-of-the-art methods fail**.
>
> Furthermore, our approach extends projected diffusion models to MRMP by introducing **two key innovations beyond Christopher et al. (2024)**:
>
>    1. We propose a relaxation of the MRMP nonconvex constraints using an Augmented Lagrangian method, transforming the problem into a convex formulation. This significantly enhances projection efficiency. In contrast, Christopher et al. (2024)'s method **faces substantial difficulties** when solving even relatively simple MRMP instances even with 3 robots due to the complexity of nonconvex projections, and thus it cannot be applicable in practice.
>
>    2. As we shown in lines 208–219 in our paper, SMD ensures the outputs generated by diffusion models can satisfy convex constraints. For a more detailed theoretical analysis, we also refer the reviewer to our response to Comment 2 from Reviewer 7Hco.
>
> Additionally, we contribute a **new benchmark** for evaluating MRMP methods, which has been valued by other reviewers and which we believe, will benefit this community. We will be more explicit about our contribution in the final version of the paper.
>
>
> **Reviewer’s Additional Points**:
>
> > The submission claims that SMD maintains feasibility as robot/obstacle density increases, however, MRMP complexity grows combinatorially with robot count
>
> We agree that MRMP is highly challenging, especially in complex environments. Recent advances in MRMP exploit diffusion models to generate diverse solutions, but existing methods fail to satisfy non-collision, which is key, in complex environments. Such issues become more pronounced as robot/obstacle density increases. As shown in our experiments, the previous state-of-the-art methods **succeeds only in empty environments with 3 robots**, yet performance **significantly degrades to a 27.2% success rate with 9 robots and 20 obstacles**.
>
> Unlike other methods whose success rates rapidly decline as robot and obstacle numbers grow, **SMD maintains feasibility in scenarios with increased robot and obstacle counts **(e.g., dense maps with 9 robots and 20 obstacles). Specifically, SMD is the only known method that **provides feasible solutions for the largest number of robots** in complex environments. Even in the most challenging dense maps, where it still maintains a near-perfect success rate.
>
> As briefly mentioned above, to mitigate the computational complexity brought by the large number of obstacles and robots, we develop an Augmented Lagrangian-based projection, which **reformulates nonconvex MRMP into a convex problem**. This significantly enhances projection efficiency and make using projection-based methods in MRMP possible, without affecting its success rate.
>
> Thank you for your points. We will further emphasize the significance of our results in the revised manuscript to help readers more directly appreciate our results.
>
> ---
>
> We would like to sincerely thank you for your insightful feedback, which is valuable to improve our manuscript. We hope our responses have addressed your concerns and we would be grateful if you could consider increasing your score to reflect this. Many thanks!

---

> > ### Comment · Reviewer_9rTn · 2025-04-02
> >
> > I thank the authors for clarifying the contribution. However, how can the satisfaction of the constraints be guaranteed theoretically? Due to the decentralized nature of multi-agent systems, I hope the authors elaborate on the applicability of SMD.

---

> > > ### Author Response · Authors · 2025-04-02
> > >
> > > We sincerely thank reviewer 9rTn for going through our response.
> > >
> > > First, let us clarify that our method, like many existing approaches to Multi-Robot Motion Planning, including its discrete counterpart (multi-agent path finding [4] ), is designed for a **centralized setting**, where a central planner coordinates all robots jointly. This is the standard assumption in prior work [1–3], and our approach follows this established line of research. This is evident from program (1) as well as the experimental setting in our paper. We also want to emphasize that our paper does not mentions the word "decentralized".
> > >
> > >
> > > 1. For **constraints feasibility guarantees**:
> > >    As explained in lines 208–219 of our paper, the derived projection operator satisfies **convex constraint feasibility guarantees**. The detailed theorem (the complete proof is in [5]) is:
> > >
> > >    *Let $\mathcal{P}\_\Omega$ be a projection onto convex set $\Omega$, $\mathbf{x}\_{t}^{i}$ be the sample at time step $t$ and iteration $i$, and $Error$ be the distance between $\mathbf{x}\_{t}^{i}$ and its nearest feasible point. Assume $\nabla\_{\mathbf{x}\_{t}} \log p(\mathbf{x}\_{t})$ is convex. For any $i \geq I$, we have:*
> > >    $$
> > >    \mathbb{E} \left[ \textit{Error}(\mathcal{U}(\mathbf{x}\_{t}^{i}), \Omega) \right] \geq \mathbb{E} \left[ \textit{Error}(\mathcal{U}(\mathcal{P}\_{\Omega}(\mathbf{x}\_{t}^{i})), \Omega) \right]
> > >    $$
> > >    *where $\mathcal{U} (\cdot)$ denotes a single update step for the sampling process.*
> > >
> > >    Although MRMP is inherently nonconvex, which prevents a direct application of this theorem, our Augmented Lagrangian-based projection reformulates MRMP into a convex problem, allowing generated trajectories from our SMD to **satisfy this relaxed version theoretically**. We handle the remaining nonconvex constraints via a dual ascent method, providing a clear stopping criterion and ensuring **constraint violations remain below the predetermined threshold**.
> > >
> > >    To the best of our knowledge, this is the first work offering these guarantees in diffusion-based trajectory planning. We will make sure to further emphasize this message and its theoretical justification in the revised manuscript.
> > >
> > > 2. The question of **decentralized multi-agent systems** is however intriguing and our method can be extended to the decentralized settings with minimal modifications.   For example, we can consider a setting where each robot knows its own goal and has access to a local environmental information within a limited sensing range. Our method can be **directly applied** to generate a collision-free trajectories that approaches the goal within this local region and run SMD for each robot iteratively. It is challenging to provide theoretical guarantees for the whole problem but our method would still ensure **constraint satisfaction** and **significantly improve efficiency** due to the reduced dimensionality of the problems. This is an exciting direction for future work!
> > >
> > > ---
> > >
> > > Thank you again for the review and suggestions. We hope that all of your questions have been addressed, and would be grateful if this could be better reflected in your final evaluation of our work. Thank you!
> > >
> > >
> > >
> > > [1] Shaoul, Yorai, et al. "Multi-robot motion planning with diffusion models."
> > >
> > > [2] Peasgood, Mike, et al. "A complete and scalable strategy for coordinating multiple robots within roadmaps."
> > >
> > > [3] van Den Berg, Jur, et al. "Centralized path planning for multiple robots: Optimal decoupling into sequential plans."
> > >
> > > [4] https://mapf.info
> > >
> > > [5] Christopher, Jacob K, et al. "Constrained synthesis with projected diffusion models."

---

### Official Review · Reviewer_7Hco · 2025-03-13

**Overall Recommendation:** 3

**Summary:**

This paper introduces Simultaneous MRMP Diffusion (SMD), a novel method for enforcing critical constraints, such as collision avoidance and kinematic feasibility, in Multi-Robot Motion Planning (MRMP). SMD integrates constrained optimization into the diffusion process to generate collision-free, kinematically feasible trajectories, thereby embedding these constraints directly within the trajectory generation pipeline. A Lagrangian-dual based approach is employed to achieve this integration. Additionally, this work presents the first benchmark for MRMP evaluation, featuring complex input maps and diverse scenarios. Experimental results demonstrate that SMD outperforms both classical and learning-based motion planners, achieving higher success rates and greater efficiency in complex multi-robot environments.

**Claims And Evidence:**

- This paper claims that diffusion processes integrated with constrained optimization helps produce collision-free, kinematically feasible trajectories but analysis on these aspects are missing from the experiments section.
- Theoretical justifications for projecting the diffusion process is unclear which leads to confusion and weakens the argument for using this particular method.

**Essential References Not Discussed:**

All necessary references are included in the paper. No further recommendations required.

**Experimental Designs Or Analyses:**

- The experiments cover various scenarios including randomly generated and structured real-world-inspired maps.
- Random maps include environments with increasing difficulty. Basic maps introduce multiple obstacles which requires navigating through narrow corridors. Dense maps contain many obstacles significantly restricting movement. The various difficulties help test the planning algorithms to their limit.
- Practical maps include corridors, warehouse storage layouts with tight aisles, rooms connected with doors. These maps are well designed to test global coordination.

**Methods And Evaluation Criteria:**

- Comparison candidates seems to be well chosen including state-of-the art methods which helps validate the results of the paper
- The random maps designed for MRMP experiments are well chosen which covers many scenarios and tests the ability of each planner to a high degree.

**Other Comments Or Suggestions:**

- On the last sentence in page 3, there is a typo: This allows it to from ~.

**Other Strengths And Weaknesses:**

- The proposed method achieves state-of-the-art results for MRMP.
- Rejection sampling or post-processing is not required for the proposed method.
- Proposing a new evaluation method is a meaningful contribution to the research community.
- Theoretical justifications lacks clarity and would benefit from further explanations.
- Figure 1's overview could be improved to clarify the projection operation.

**Questions For Authors:**

- Does projection actually help guarantee collision avoidance and kinematic feasibility?
- How does Simultaneous MRMP Diffusion perform on more complex plans where the trajectories are more complex?

**Relation To Broader Scientific Literature:**

- This paper proposes to constrain the diffusion process to avoid collisions and satisfy kinematic feasibility. Compared to previous works, a novel constrained optimization process is integrated into the diffusion process.
- Furthermore, this paper proposes a method for evaluating Multi-Robot Motion planning (MRMP) which can be used for future works.

**Theoretical Claims:**

- How the equations constrains each robot's trajectory is rather unclear and requires further explanation.
- In section 4.2 the process of the inequality being rewritten as equalities are missing which makes it harder to understand and appreciate.

---

> ### Author Rebuttal · Authors · 2025-03-31
>
> Thank you for the helpful comments, in particular the acknowledgement of the **strong empirical performance** of our proposed SMD and **a novel benchmark** for MRMP. We provide below our answers to your constructive questions.
>
> - **Q1: Does projection actually help guarantee collision avoidance and kinematic feasibility?**
>
> From the **theoretical perspective**:
>
> Firstly, In lines 208–219, we demonstrate that the derived projeciton operator satisfies both the **convergence and convex constraint feasibility guarantees**. Although MRMP is inherently nonconvex, our Augmented Lagrangian-based projection reformulates it into a convex problem, allowing generated trajectories from our SMD to **satisfy relaxed MRMP constraints theoretically**. We address remaining nonconvex constraints via a dual ascent method, providing a clear stopping criterion and ensuring **constraint violations less than a predetermined threshold**. To the best of our knowledge, this is the first work offering these guarantees in diffusion-based trajectory planning.
>
> Next, if you ask *why project throughout the diffusion process?* The shortcomings of post-processing methods are well documented within the literature: It is shown to substantially degrade sample quality ([1] highlighs that samples are often too "implausible to be corrected... [and] may be pushed away from the data distribution.").
>
> From the **empirical perspective**:
>
> We do provide this analysis in our experiments. Performance is measured by success rate: **the percentage of collision-free, kinematically feasible trajectories in test cases**. SMD consistently achieves the highest success rate, significantly outperforming other diffusion-based approaches. Specifically, the **key difference between SMD and MPD (a baseline method in our experiments) is integrating constrained optimization into the diffusion process**. We will further clarify this aspect in the revised version providing a numerical summary in the introduction.
>
> We will be happy to include a more formal statement of theoretical analysis in the final version.
>
> - **Q2: How does Simultaneous MRMP Diffusion perform on more complex plans where the trajectories are more complex?**
>
> We have evaluated our method in scenarios **significantly more challenging than those explored in previous studies**. Compared to  MPD, we extend our experiments to multi-robot systems; compared to MMD (the previous SOTA appraoch, which was deposited on ArXiv only 3 months before this submission), we increase both the number and density of obstacles. **In fact, our paper even proposes a new benchmark that provides the highest degree of scaling within Multi-Robot Motion Planning to date.** Our method consistently achieves superior performance under these complex conditions. We agree that further investigation into more complicated scenarios, such as 3D environments, would be valuable; This is an important direction for future work.
>
> **Reviewer’s Additional Points**:
>
> > **A1**: Feasibility analysis in experiments section & Theoretical justifications for projecting the diffusion process
>
> Please refer to our response to Q1 for details.
>
> > **A2**: The process and effectiveness of reformulating inequality constraints into equality ones
>
> We take a general inequality constraints as an example:
> $$
> h(x)\geq b,
> $$
> where $x$ is the decision variable and $b$ is a parameter. Directly handling Eq (1) with Augmented Lagrangian methods are complicated because multipliers for inequality constraints should be larger than 0, which introduces extra decision logic.
>
> Instead, equality constraints can lead better convergence properties and simplify multiplier updates:
> $$
> \boldsymbol{\nu}^{k+1}=\boldsymbol{\nu}^{k}+\alpha^k\boldsymbol{h}(x^k)
> $$
> where $\boldsymbol{\nu}$ represents the multiplier and $\alpha$ is the step size.
> To transform inequalities into equalities, we introduce non-negative auxiliary variables $s$:
> $$
> h(x)-b-s=0.
> $$
> The presence of non-negative variable $s$ can ensure Eq. (3) is equivalent to Eq. (1):
>
> - If $h(x)\geq b$ , then $s=h(x)-b\geq0$ makes the equality hold.
> - If $h(x)-b-s=0$ and $s\geq0$, then $h(x)-b=s\geq0$.
>
> We will provide a more detailed description in the revised paper.
>
> > **A3**: Visualizations for the Benchmark
>
> We will include linked a GitHub page with visualizations in the final version:  https://drive.google.com/drive/folders/1ZdfNmA9BfIdshIbRN3AmAcyvXakhrhkr?usp=drive_link
>
> > **A4**: Figure 1’s overview
>
> A new figure is here: https://drive.google.com/file/d/1W8tuHwVLBV-2zZ9It0pLIdny1ZHchoxF/view?usp=drive_link
>
> Could you let us know if you have further suggestions?
>
> ---
>
> We want to thank the reviewer again for their valuable feedback. We believe that our response has addressed each of your comments but are happy to provide additional details if requested. We would be grateful if you could consider increasing your score to reflect this. Many thanks!
>
>
>
> [1] Yuan et al. "Physdiff: Physics-guided human motion diffusion model."

---

> > ### Comment · Reviewer_7Hco · 2025-04-02
> >
> > The authors provided sincere and well-reasoned responses to the reviewer’s questions, supported by both theoretical explanations and experimental evidence. In particular, they adequately addressed feedback regarding the convergence and constraint satisfaction of the projection method, superior performance in complex scenarios, the transformation of mathematical constraints, and improvements to visual materials. They also clearly outlined their revision plans. However, it is somewhat regrettable that the theoretical justification for the projection and quantitative details on the constraint violation tolerance were not directly included in the main paper. I hope all of the authors' responses will be thoroughly reflected in the revised manuscript.

---

> > > ### Author Response · Authors · 2025-04-02
> > >
> > > We sincerely thank the reviewer 7Hco for the encouraging feedback! Of course, we will make sure to incorporate our responses into the final version, including the formal theoretical justification of the projection method and the quantitative details on constraint violation tolerance. Indeed, we have already modified the paper on our end.
> > >
> > > If there are no further points of clarification, we would be grateful if you could consider further championing this work to reflect the improvements and clarifications we have provided. Thank you!

---

### Decision · Program_Chairs · 2025-05-01

**Decision:**

Accept (poster)

**Comment:**

This work proposes Simultaneous Multi-robot motion planning Diffusion (SMD) for the problem of finding conflict-free trajectories under kinematic constraints.

The reviewers agree that the proposed approach is an interesting and promising application of diffusion models in the multi-robot navigation context. There are some shared concerns about a straightforward centralised architecture that does not enable scaling to larger teams of robots; and some points regarding clarity of presentation that should be addressed.

However, all-in-all there is an agreement that presenting this paper is worthy of being presented at ICML.